# Identification of rare de novo epigenetic variations in congenital disorders

Mafalda Barbosa[1,2,3], Ricky S. Joshi[1], Paras Garg[1], Alejandro Martin-Trujillo[1], Nihir Patel [1], Bharati Jadhav[1], Corey T. Watson[1], William Gibson[1], Kelsey Chetnik[4], Chloe Tessereau[1], Hui Mei[5,14], Silvia De Rubeis[3,6], Jennifer Reichert[3,6], Fatima Lopes[7], Lisenka E.L.M. Vissers[8], Tjitske Kleefstra[8], Dorothy E. Grice[9,10], Lisa Edelmann[5], Gabriela Soares[11], Patricia Maciel[7], Han G. Brunner[8,12], Joseph D. Buxbaum[1,3,6,10], Bruce D. Gelb [1,13] & Andrew J. Sharp[1,2]

Certain human traits such as neurodevelopmental disorders (NDs) and congenital anomalies (CAs) are believed to be primarily genetic in origin. However, even after whole-genome sequencing (WGS), a substantial fraction of such disorders remain unexplained. We hypothesize that some cases of ND–CA are caused by aberrant DNA methylation leading to dysregulated genome function. Comparing DNA methylation profiles from 489 individuals with ND–CAs against 1534 controls, we identify epivariations as a frequent occurrence in the human genome. De novo epivariations are significantly enriched in cases, while RNAseq analysis shows that epivariations often have an impact on gene expression comparable to loss-of-function mutations. Additionally, we detect and replicate an enrichment of rare sequence mutations overlapping CTCF binding sites close to epivariations, providing a rationale for interpreting non-coding variation. We propose that epivariations contribute to the pathogenesis of some patients with unexplained ND–CAs, and as such likely have diagnostic relevance.

[1] The Mindich Child Health & Development Institute and the Department of Genetics & Genomic Sciences, Icahn School of Medicine at Mount Sinai, New York, NY 10029, USA. [2] Graduate School of Biomedical Sciences, Icahn School of Medicine at Mount Sinai, New York, NY 10029, USA. [3] The Seaver Autism Center for Research and Treatment, Icahn School of Medicine at Mount Sinai, New York, NY 10029, USA. [4] School of Theoretical and Applied Sciences, Ramapo College of New Jersey, Mahwah, NJ 07430, USA. [5] Department of Genetics and Genomic Sciences, Icahn Institute for Genomics and Multiscale Biology, Icahn School of Medicine at Mount Sinai, New York, NY 10029, USA. [6] Department of Psychiatry, Icahn School of Medicine at Mount Sinai, New York, NY 10029, USA. [7] ICVS/3B's PT Government Associate Laboratory, Life and Health Sciences Research Institute, School of Medicine, University of Minho, Braga/Guimarães 4710-057, Portugal. [8] Radboud University Medical Center, Department of Human Genetics, Donders Institute for Brain, Cognition and Behaviour, Nijmegen 6500 HB, The Netherlands. [9] The Division of Tics, OCD and Related Disorders, Department of Psychiatry, and Mindich Child Health and Development Institute, Icahn School of Medicine at Mount Sinai, New York, NY 10029, USA. [10] The Friedman Brain Institute, Icahn School of Medicine at Mount Sinai, New York, NY 10029, USA. [11] Center for Medical Genetics Dr. Jacinto Magalhães, Porto Hospital Center, Porto 4050-106, Portugal. [12] Maastricht University Medical Center, Department of Clinical Genetics, GROW School for Oncology and Developmental Biology, Maastricht 6229 HX, The Netherlands. [13] Department of Pediatrics, Icahn School of Medicine at Mount Sinai, New York, NY 10029, USA. [14] Cardiogenetic Program, GeneDx, Inc., Gaithersburg, MD 20877, USA. These authors contributed equally: Mafalda Barbosa, Ricky S. Joshi, Paras Garg. Correspondence and requests for materials should be addressed to A.J.S. (email: andrew.sharp@mssm.edu)

Epimutations represent a class of mutational event where the epigenetic status of a genomic locus deviates significantly from the normal state, and can be classified into two main types: primary epimutations are thought to represent stochastic errors in the establishment or maintenance of an epigenetic state, while secondary epimutations are downstream events related to an underlying change in the DNA sequence[1]. Both secondary and primary epimutations that originate in the germline will be constitutive events found in all cells. In contrast, primary epimutations that occur post-fertilization may result in somatic mosaicism. Constitutive (i.e., non-mosaic) epimutations are known to underlie several genetic disorders that can be identified in blood-derived DNA: 5–15% of patients with hereditary non-polyposis colon cancer present with constitutional *MLH1* promoter methylation[2], and fragile X syndrome, the most common cause of inherited intellectual disability, results from a secondary epimutation in which hypermethylation of an expanded CGG repeat at the *FMR1* promoter causes transcriptional silencing[3].

With the recent dramatic advances in genomic technologies, genome-wide surveys of cohorts of patients with neurodevelopmental disorders (NDs) and congenital anomalies (CAs) (ND–CAs) for point mutations and structural variations have greatly advanced our understanding of their genetic etiologies[4, 5]. However, even after whole genome sequencing (WGS), no causative mutation can be identified in many such cases[6]. We hypothesized that some cases of ND–CA that remain refractory to conventional sequence-based analysis harbor rare epigenetic aberrations (termed epivariations), which are associated with dysregulation of normal genome function, and that these would be missed by the conventional sequencing approaches. We identify rare epigenetic changes that are absent in thousands of controls in ~20% of patients with ND–CA. From large-scale sequencing, population and expression studies, we conclude that epivariations are: (i) frequently associated with extreme outlier and mono-allelic gene expression; (ii) generally conserved across multiple tissues within an individual, validating the use of blood DNA to study ND–CA; (iii) sometimes occur secondary to *cis*-linked regulatory mutations, providing a rationale for interpreting non-coding genetic variants; (iv) can occur sporadically with a remarkably high de novo rate, and (v) a subset exhibit non-Mendelian inheritance, suggesting they are often being reset between generations by epigenetic reprogramming. We propose that epivariations likely contribute to the pathogenesis of some patients with unexplained ND–CAs, and suggest that epigenome profiling represents a promising method for the study of human disease that complements sequence-based approaches.

## Results

**Identification of epivariations in cases and controls.** We studied a cohort comprising 489 individuals with ND–CA: most had been previously tested by copy number variation (CNV) microarray, all had undergone exome sequencing, and some had undergone WGS, yet no putatively pathogenic mutations had been identified. Almost 90% of the patients had an ND, 50% were classified as having an autism spectrum disorder, 16% had an epilepsy/seizure phenotype; 65% also had multiple CA, including congenital heart defects (CHD) (36%), facial dysmorphisms (29%), growth anomalies (22%), and micro/macrocephaly (13%) (full details in Supplementary Data 1). We hypothesized that this cohort represented an optimal population in which to search for novel pathogenic epivariations since an underlying genomic abnormality was suspected, but many common environmental and genetic causes of ND–CA had been excluded. Methylation profiling in ND–CA samples was performed with the Illumina Infinium Human Methylation 450 BeadChip (450k array). Profiles in

each ND–CA sample were compared individually against a control cohort comprising 1534 unrelated individuals from four publicly available datasets (GSE36064, GSE40279, GSE42861, and GSE53045). We also searched for epivariations in two cohorts of population controls by comparison against this same set of 1534 individuals: 117 families (GSE56105)[7] were used to assess the inheritance of epivariations in controls (Supplementary Data 2); 2711 unrelated individuals (GSE55763)[8] were used to assess the frequency of epivariations in the general population (Supplementary Data 3). We utilized a sliding window approach to identify epivariations in each sample, defined as 1 kb regions containing ≥3 probes showing rare outlier methylation absent in the set of 1534 common control individuals (see Supplementary Fig. 1 and Methods section). After stringent quality control, including removal of loci with clusters of poorly hybridizing probes and extensive manual curation to remove technical and batch effects, we identified a total of 143 epivariations in 114 ND–CA samples (i.e., 23% of the probands tested). Twenty percent of the ND–CA cohort carried one epivariation ($n = 98$), while 3% of the individuals tested presented two or more epivariations ($n = 16$) (Supplementary Data 4 and Supplementary Fig. 2).

Using PCR/bisulfite sequencing, we attempted orthogonal confirmation for 70 epivariations. We observed concordant changes in methylation for 55 of the 58 assays that provided useful data, yielding a 95% true positive rate for differentially methylated regions (DMRs) detected by array (Supplementary Data 5). Allelic analysis demonstrated that these epivariations represent large methylation changes specifically on one allele, consistent with the hypothesis that epivariations represent allelic events. In most cases, we observed two clusters of largely methylated and unmethylated reads occurring in approximately equal proportions (Fig. 1), although in some instances the interpretation of validation experiments was made complex due to highly biased allelic representation, presumably reflecting preferential PCR amplification of one allele (Supplementary Fig. 3).

In addition to searching for epivariations in samples with ND–CA, we also screened two large cohorts of population controls, identifying a total of 719 DMRs in the 3326 control samples analyzed (Supplementary Data 2 and 3). Thus, epivariations are a relatively common occurrence in the human genome, and are not always associated with any discernable clinical phenotype. Twenty-four of the epivariations identified in our cases with ND–CA were also found in one or more of these controls, therefore indicating that either these DMRs are unrelated to the patient phenotype, or perhaps are associated with incomplete penetrance. However, we observed a 1.2-fold enrichment in the frequency of epivariations in the 489 ND–CA samples when compared to 2711 population controls (Supplementary Fig. 2), although this does not reach statistical significance ($p = 0.058$, two-sided Fisher's exact test).

Using a combination of 450k arrays and bisulfite PCR/sequencing assays, we were able to assess the inheritance of 57 DMRs identified in our patients with ND–CA: 33 of the 57 epivariations tested were also present in apparently unaffected parents, and thus represent inherited events. However, 42% ($n = 24$) of the epivariations identified in patients with ND–CA were absent in both parental samples, and thus occurred as de novo events. When compared to epivariations found in 117 control pedigrees[7] (Supplementary Data 2), this represents a 2.8-fold enrichment in the rate of de novo epivariations in cases compared to controls ($p = 0.007$, two-sided Fisher's exact test) (Fig. 2). Thus, while it is currently unclear whether many of the epivariations identified contribute to the phenotypes of the patients in our study, the paradigm of de novo mutational events

echoes that observed for other classes of genetic mutation (copy number and single nucleotide variation (SNV)) deemed pathogenic in ND and CHD cohorts[9, 10].

In addition to their de novo nature, recurrence of mutations found in unrelated patients with a similar phenotype is commonly used as a way of assigning significant evidence for the involvement of a specific gene or locus in disease. We identified 12 recurrent epivariations (Supplementary Fig. 4), i.e., the same methylation change was identified in multiple unrelated probands. Of these, two epivariations encompassed the promoters of genes known to show altered methylation in congenital disease (MEG3 and FMR1)[6, 11], showing that our approach successfully

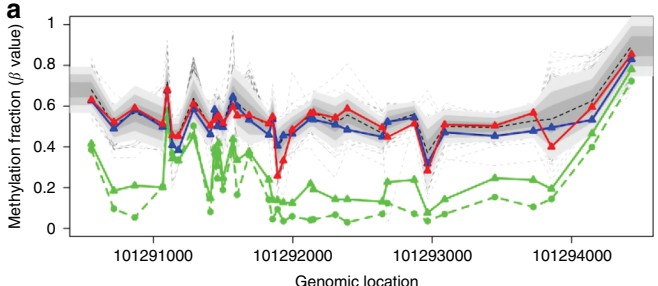

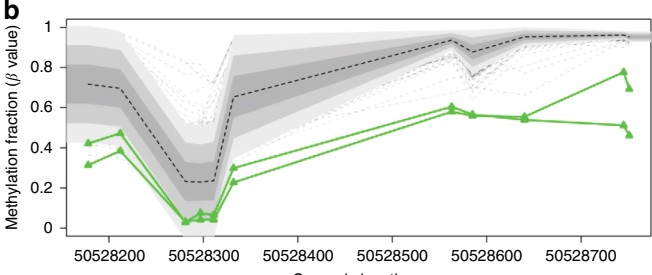

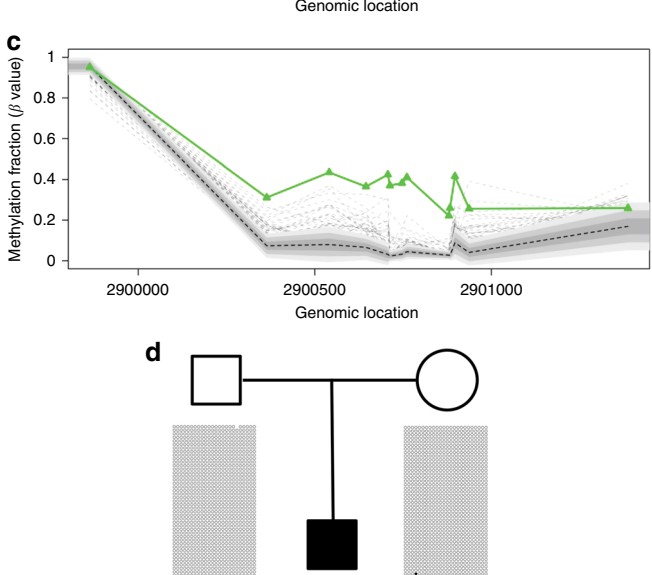

detects pathogenic epivariations. The two males identified with hypermethylation at FMR1 had phenotypes consistent with a diagnosis of fragile X, primarily intellectual disability (ID) and behavioral anomalies. While both had previously been tested by PCR and reported as normal, subsequent Southern blot testing confirmed the presence of the classical FMR1 triplet repeat expansion, although in one case this was an apparent mosaic event. A third recurrent epivariation coincides with a fragile site containing a hypermethylated triplet repeat expansion (FRA10AC1)[12] although this, and four other recurrent epivariations detected in our disease cohort, was also identified in population controls, suggesting that they are unlikely to be pathogenic. One of the novel recurrent epivariations detected only in our patient cohort was found in two patients with CHD (Probands 22 and 117), and represents a recurrent hypomethylation defect at the promoter—5′ UTR—first exon of MOV10L1, a gene with an embryonic heart-specific isoform that interacts with the master cardiac transcription factor NKX2.5[13] (Fig. 1). One patient with this epivariation at MOV10L1 presented with double outlet right ventricle, hypoplastic left ventricle, asplenia, and short stature, while the second presented with pulmonary stenosis, laryngo-bronchio-tracheomalacia, and foot polydactyly. Finally, using less stringent criteria for identifying DMRs (see Methods), we detected methylation defects in 11 probands at 10 imprinted loci[14] (Supplementary Fig. 5 and Supplementary Table 1), 90% of which occurred de novo. Of note, we observed loss of methylation at two known imprinted loci that have no prior disease associations (NAA60/ZNF597 in Probands 6 and 62, and L3MBTL1 in Proband 308), although in both cases similar losses of methylation were also observed in population controls, making the pathogenic significance of loss of imprinting at these loci unclear.

**Regulatory mutations underlie some epivariations.** Based on previous studies[15, 16], we hypothesized that some epivariations might occur secondarily to an underlying regulatory sequence mutation. In order to identify mutations disrupting regulatory elements (e.g., transcription factor binding sites) that might underlie the methylation changes observed in our cohort, we performed high-resolution array comparative genomic hybridization (CGH) and targeted DNA sequencing of 50 DMRs and their flanking sequences. We detected rare sequence mutations

**Fig. 1** Large gains and losses of DNA methylation identified in patients with ND-CA. Plots **a**, **b**, and **c** show β values obtained from Illumina 450k array for probands (highlighted in green) and 1534 controls (shades of gray corresponding to ±1, ±1.5, and ±2 standard deviations from the population mean, represented by the dashed black line; dashed gray lines represent controls with outlier methylation levels). **a** Recurrent hypomethylation of the imprinted locus of MEG3 (hg19: chr14:101290194–101294429) in Proband 398 (solid green line) and Proband 146 (dashed green line). The epivariation in Proband 398 is de novo as both mother (red line) and father (blue line) present methylation profiles similar to controls. **b** Recurrent hypomethylation at the promoter, 5′ UTR, and first exon of MOV10L1 (hg19: chr22:50528178–50528751) observed in two unrelated probands: Proband 22 (de novo epivariation) and Proband 117 (inheritance unknown). **c** Hypermethylation of ZNF57 in Proband 381. **d** Pedigree and graphical representation of the methyl-seq data consistent with allele-specific nature of a de novo hypermethylation identified in ZNF57 is shown. Each plot shows the methylation pattern for an amplicon, with each row representing a single bisulfite read and each column one CpG in the amplicon. Black circles are methylated CpGs and white circles unmethylated CpGs. Based on the presence of a heterozygous SNP within the DMR (hg19: chr19:2900643), the observed gain of methylation occurs specifically on one allele: each pie chart shows the methylated fraction of reads per CpG

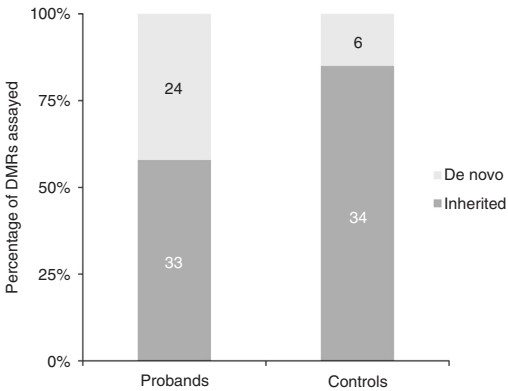

**Fig. 2** A significant excess of de novo epivariations found in patients with ND–CA. We observed a 2.8-fold enrichment for de novo epivariations in cases (n = 24 out of 57) when compared to controls (n = 6 out of 40) (p = 0.007, two-sided Fisher's exact test)

that co-segregated with epivariations and potentially impact regulatory elements at 24% of the loci tested: six CNVs (Fig. 3 and Supplementary Fig. 6) and seven SNVs (Supplementary Data 4, 6, and 7). Where inheritance data from parental samples were available, we found that all of these rare CNVs and SNVs segregated with the presence of the DMR, suggesting that the epivariations occurred secondarily to the underlying sequence mutation.

Of the rare segregating SNVs detected at DMRs, three were SNVs within the canonical binding sites for CTCF (CCCTC-binding factor), a transcription factor with roles in chromatin organization (Fig. 4), including a de novo SNV that disrupts a CTCF binding motif in association with a de novo epivariation (Proband 70) (Supplementary Fig. 7). In each case, the disrupted CTCF motif was either overlapping or very close to (separation <1 kb) the DMR. This represents a significant enrichment for rare SNVs disrupting CTCF binding sites in the vicinity of epivariations when compared to the same regions in other samples in whom we performed targeted sequencing, but who did not carry epivariations at these loci (p = 0.0015, two-sided Fisher's exact test), strongly implicating rare cis-linked variants in regulatory sequence as a causative factor underlying some epivariations. Furthermore, given the low frequency of de novo SNVs and epivariations in the genome, it is highly unlikely that a de novo SNV and a de novo epivariation would co-occur at the same locus in an individual by chance, providing additional support that some epivariations represent secondary events caused by disruption of CTCF binding. Using paired methylation and sequence data from 90 individuals studied by the 1000 Genomes Project (Supplementary Data 8), we replicated this enrichment for rare SNVs disrupting CTCF binding motifs around epivariations (p = 0.049, two-sided Fisher's exact test), identifying two rare CTCF-disrupting SNVs, one of which co-segregates with the presence of an epivariation in multiple unrelated individuals. Though readily detectable by WGS, there is considerable difficulty in interpreting the functional significance of variants outside of coding regions. Thus, we propose that the use of epigenome profiling represents a complementary approach that can provide a rationale for interpreting non-coding genetic variation.

**Functional consequences of epivariations**. In order to provide insight into the biology and functional consequences of epivariations[17], we performed studies of gene expression, inheritance, and tissue conservation using datasets of DNA methylation (Supplementary Data 9), gene expression (Supplementary

Data 10), and genotype data derived from population controls[18–21]. Using paired RNAseq and DNA methylation data in 90 samples from the 1000 Genomes Project, we verified that epivariations encompassing gene promoters were often associated with large changes in gene expression, with hypomethylation leading to increased expression and hypermethylation to transcriptional repression, consistent with the known repressive effects of promoter DNA methylation (p = 9.2 × 10⁻⁵, Wilcoxon Rank-Sum test) (Fig. 5, Supplementary Data 10)[22]. We also observed that many hypermethylated epivariations at promoters are associated with complete silencing of one allele (Supplementary Fig. 8). While these observations were made in a control cohort, this suggests that some epivariations have an impact comparable to that of loss-of-function coding mutations.

**Epivariations are generally present in multiple tissues**. While epigenetic profiles can vary substantially between cell types[23], it is unclear whether similar cell-specific variability exists for epivariations. To address that, we analyzed cohorts in which methylation profiles were available from multiple different tissues[21]. In samples from the GenCord population, in which methylation data from fibroblasts, B cells, and T cells sampled from dozens of newborns are available, by first identifying DMRs in T cells, we observed a very strong concordance for outlier methylation at the same locus in fibroblasts derived from the same individual (Spearman rank correlation of 0.75, p = 1.2 × 10⁻²⁷, Wilcoxon Rank-Sum test) (Fig. 6, Supplementary Data 9). Similar concordance for outlier methylation at epivariations was also observed between fibroblasts and B cells.

A similar trend for conservation of epivariations across multiple different post-mortem tissues was also observed in a second cohort[24]. Here, epivariations found in blood were nearly all visible in multiple other somatic tissues sampled from the same individual (Supplementary Fig. 9). Thus, we conclude that the majority of epivariations are constitutive events found in multiple tissues. This provides confidence that epivariations of relevance for ND–CA can be detected using DNA extracted from readily available sources such as peripheral blood leukocytes.

**Evidence for non-Mendelian inheritance of epivariations**. Despite strong evidence that some of the epivariations we observed are secondary events related to the presence of an underlying sequence change (Figs. 3 and 4), we were unable to detect cis-linked sequence mutations associated with the majority of epivariations in our cohort, suggesting that these might instead represent primary epivariations that arose sporadically. As the mammalian genome undergoes several rounds of demethylation and remethylation during gametogenesis, embryonic and somatic development[25], theoretically there is considerable potential for primary epivariations to be reset to the default state. We therefore assessed how often epivariations are stably transmitted between parents and their offspring. Using a large control cohort comprising 117 nuclear families[7], we studied the heritability of epivariations between generations, identifying 47 epivariations segregating within these pedigrees. We observed a marked deviation from the expectations of Mendelian inheritance, with only 32 instances of parent–child transmission in 95 informative meioses; significantly fewer than the Mendelian expectation of 47.5 transmissions (p = 0.027, two-sided Fisher's exact test) (Supplementary Data 2). Therefore, this apparent reduction in heritability indicates that primary epivariations often exhibit non-Mendelian inheritance, and suggests they are frequently reset between generations by epigenetic reprogramming[26–28].

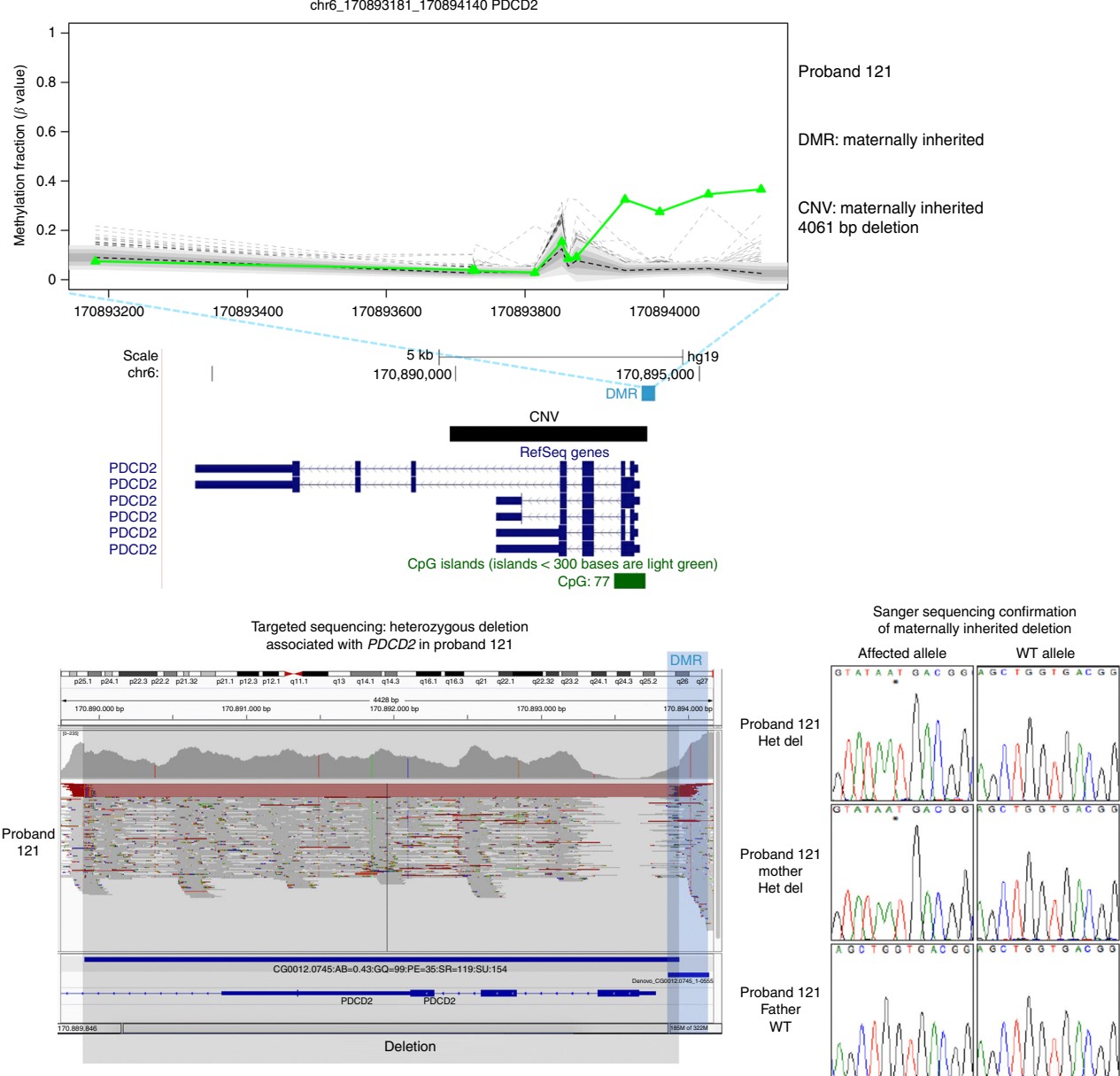

**Fig. 3** Detection of rare CNVs by targeted sequencing of epivariations and their flanks. Proband 121 carries a maternally inherited DMR at the *PDCD2* locus. We identified a maternally inherited heterozygous 4061 bp deletion flanking the DMR. Two other similar examples are shown in Supplementary Fig. 6

## Discussion

In this study, we set out to investigate the prevalence, causes, and consequences of epigenetic defects in the human genome, and to study their potential role in the etiology of ND–CA disorders. By performing epigenome profiling in a large cohort of 489 patients with diverse ND–CA, all of whom had previously undergone microarray testing and/or exome or genome sequencing, in addition to analyzing >5000 population controls, we demonstrate that epivariations are a relatively frequent occurrence in the human population. Subsequent analysis of cell lines showed that the presence of epivariations is often associated with large changes in the expression of *cis*-linked genes, indicating functional consequences on the genome. Furthermore, we demonstrated that epivariations are generally conserved across multiple tissues, validating the use of peripheral blood for the studies of ND–CA.

We observed both recurrent epivariations in cases that were absent in thousands of population controls, and a significant excess of de novo epivariations in cases compared to controls, both of which are hallmarks often associated with pathogenic variants. Despite this, the pathogenic significance of many of the epivariations we identified remains uncertain. For example, in many cases, epivariations were inherited from apparently unaffected parents, suggesting that they are unlinked to the observed phenotype. While it is likely that the presence of some epivariations we observed in ND–CA cases is unrelated to patient phenotype, we note that not all inherited events are benign, and there are many examples of rare inherited sequence variants that show variable penetrance[29].

Studies of the inheritance of epivariations in families showed that they can occur de novo at very high frequency (up to 42%), yet also show significantly reduced heritability compared to Mendelian expectations, suggesting that they are often reset during meiosis. However, in contrast to this dynamic process of frequent gain and loss, targeted sequencing and array CGH identified segregating rare sequence variants that disrupt

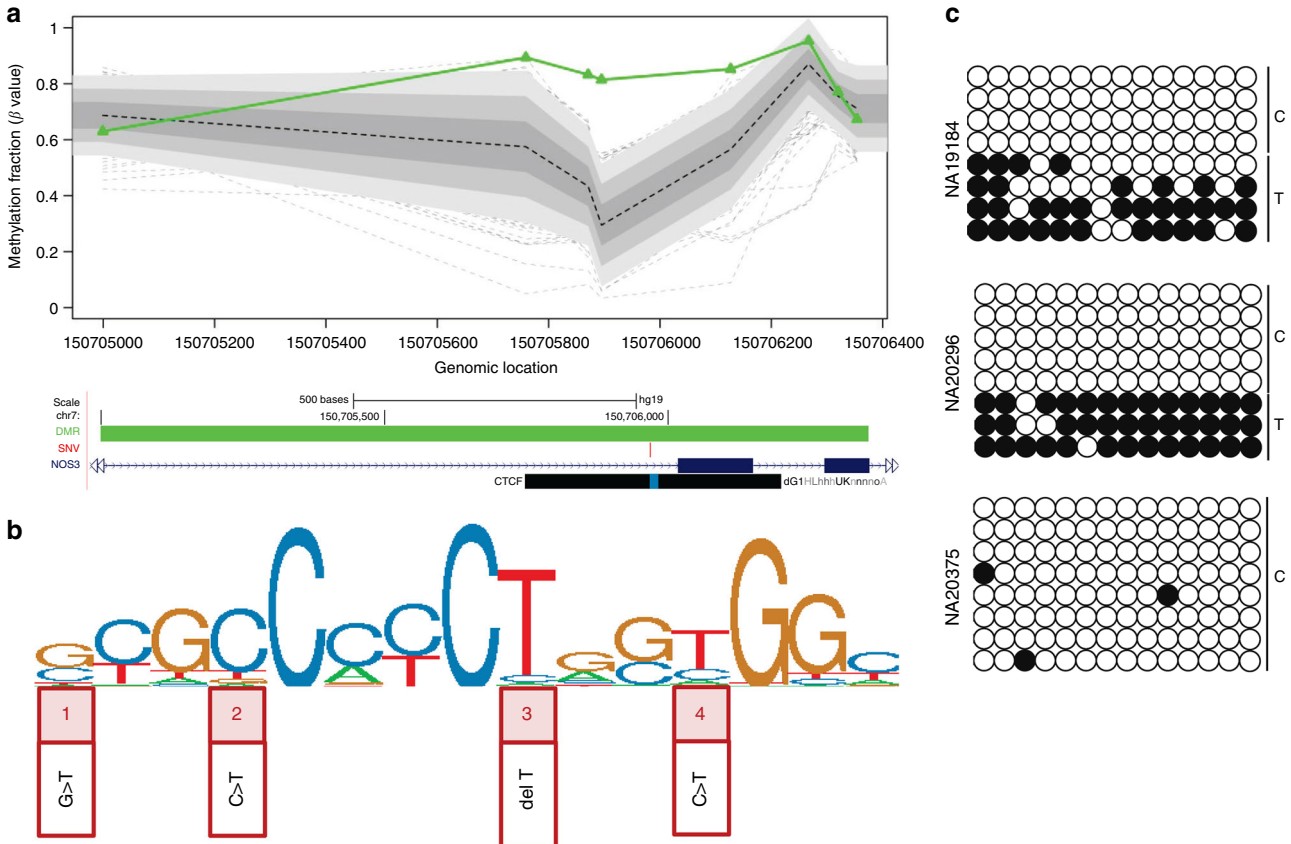

**Fig. 4** Targeted sequencing of epivariation loci identifies a significant enrichment of rare SNVs within the CTCF canonical binding motif.
**a** Hypermethylation in *NOS3* (chr7:150704999-150706354) in Proband 103 (outlier in green); in the lower UCSC Genome Browser view, the DMR location is shown as a green bar, and a rare SNV that lies within the CTCF binding motif (blue region within black bar) in this same individual is shown in red. **b** CTCF motif according to ENCODE Factorbook repository. Rare SNVs overlapping this CTCF binding motif were identified in four DMR carriers: (1) Proband 103: SNV (chr7:150705968 G>T), (2) Proband 70: SNV (chr19:295321 C>T), (3) Proband 176: 1 bp deletion (chr20:36793857 delT), (4) HapMap samples NA19239, NA19184; NA20296: rs116767319 (chr5:177707147 C>T). **c** 450k array analysis identified a DMR in NA19239, and a rare SNV (rs116767319) within a CTCF-binding motif in *cis*. We tested two other carriers of rs116767319 (NA19184 and NA20296) using allele-specific bisulfite sequencing, and found that both showed methylation on the T allele, thus confirming segregation of the epivariation with SNV. In contrast, a sample (NA20375) homozygous for the reference C allele is unmethylated

annotated regulatory elements associated with 24% of the epi-variations we investigated, indicating that a subset are likely secondary events caused by underlying sequence variations. Thus our data are consistent with a model in which some epivariations are primary events that occur sporadically, but are often reset during the waves of epigenome remodeling that occur during meiosis and early embryonic development, while others are secondary events that occur as a result of *cis*-linked regulatory sequence mutations. Consistent with this, previous studies in humans[27] and mouse[26, 28] have shown that primary epimutations are often reset during meiosis, while secondary epivariations have been observed to remain stable through multiple generations[30].

In addition to mutations that disrupt regulatory elements such as TF binding sites, expansions of GC-rich tandem repeats can also result in local DNA hypermethylation. Indeed, we identified multiple individuals with gains of methylation at known CGG repeats, including *FMR1/FRAXA*[3] (two cases), *XYLT1/FRA16A* (one case)[31], *FRA10AC1/FRA10* (two cases and nine controls)[32], and *DIP2B/FRA12* (one control)[33]. Both individuals with *FMR1* hypermethylation were shown to carry the classic CGG expansion that causes Fragile X, and although not tested, it is likely that the gains of methylation observed at the other three known fragile sites are also caused by similar repeat expansions. While our targeted sequencing experiments of other epivariations did not

identify any novel tandem repeat expansions, this class of mutation is difficult to detect with short-read sequencing. Thus, it remains possible that some of the hypermethylated epivariations we observed might be caused by this mechanism.

Previous studies have made attempts to investigate the prevalence of epigenetic changes in patients with ND–CA. Kolarova et al.[34] compared the DNA methylation profiles of 82 patients against 19 controls, identifying a total of 157 DMRs. Consistent with our own findings, Kolarova et al.[34] also identified patients with hypomethylation defects of *MEG3* and *MOV10L1*. While loss of imprinting at *MEG3* is a known cause of Temple syndrome, the recurrent observation of hypomethylation of *MOV10L1* is significant (total $n = 3$ of 571 cases versus 0 of 4878 controls, $p = 0.001$, Fisher's Exact test), and provides additional evidence implicating this locus with developmental defects. Similarly Aref-Eshghi et al.[35] assessed methylation patterns in a total of 528 samples, identifying altered methylation levels at several imprinted loci that were absent in controls. Consistent with our own study, their cohort included two patients with hypermethylation of *HM13*, thus demonstrating this as another recurrent alteration found in patients with ND–CA, and suggesting that this may be an imprinting disorder.

Our study shows that epivariations are a relatively common feature in the human genome, that some are associated with

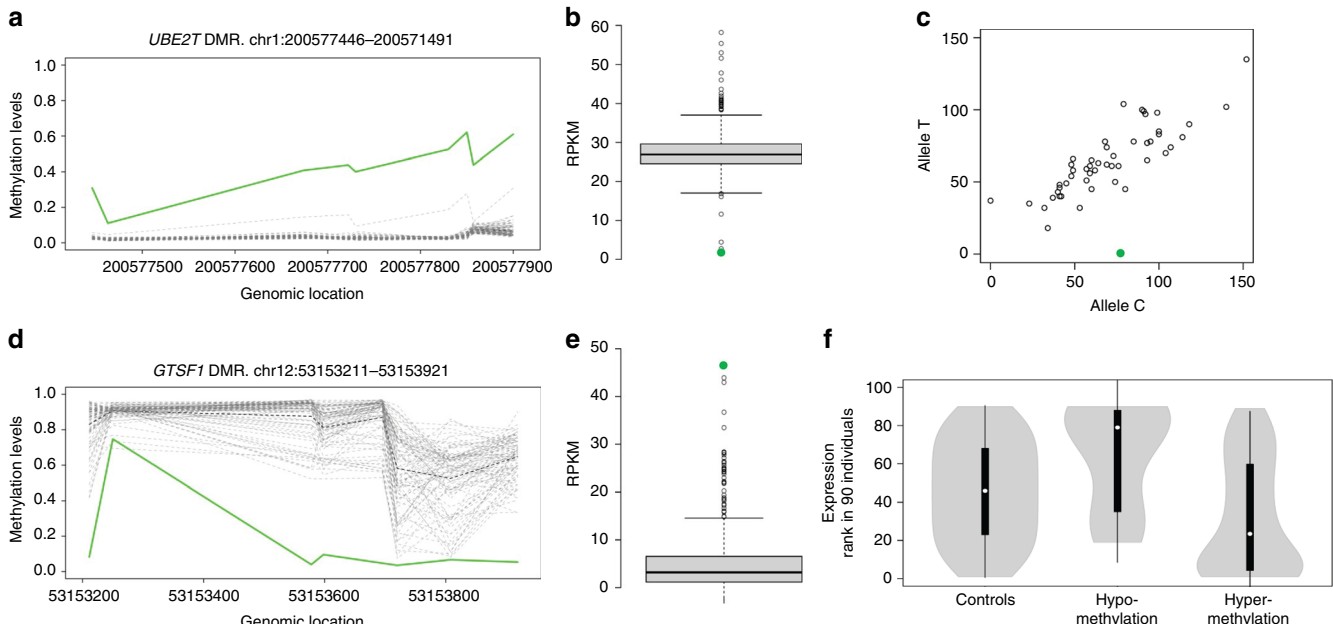

**Fig. 5** Epivariations are frequently associated with large changes in gene expression. We identified epivariations in 90 lymphoblastoid cell lines studied as part of the 1000 Genomes Project, and combined these with SNP genotypes and RNAseq data from a total of 462 samples to measure quantitative and allelic effects of epivariations on gene expression. **a** An individual with hypermethylation of the *UBE2T* promoter (solid green line) compared to 89 other individuals (dashed gray lines) presented **b** the lowest gene expression (green dot on the boxplot) of the cohort. **c** Using heterozygous SNPs within RNAseq reads, we observed monoallelic expression of *UBE2T* in the epivariation carrier (outlier highlighted in green). **d** An individual with hypomethylation of the *GTSF1* promoter (solid green line) presents **e** the highest level of expression (green dot on the boxplot). **f** Violin plots show that individuals with hypomethylated epivariations at gene promoters show significantly increased expression of that gene, whereas individuals with hypermethylated promoter epivariations show significantly reduced expression of that gene ($p = 9.2 \times 10^{-5}$, Wilcoxon Rank-Sum test). In box plots (**b** and **e**), the center line shows the median; box limits indicate the 25th and 75th percentiles; whiskers extend 1.5 times the interquartile range from the 25th and 75th percentiles; outliers are shown as individual points. In the violin plot (**f**), the white dots show the median; box limits indicate the 25th and 75th percentiles; whiskers extend 1.5 times the interquartile range from the 25th and 75th percentiles

changes in the local gene expression, and raises the possibility that they may be implicated in the etiology of developmental disorders. In an era when WGS is being applied to many thousands of human genomes, epivariations represent a class of genetic variation that remains undetectable by purely sequence-based approaches. We anticipate that future studies exploring the relationship between sequence variation and epigenetic state will further illuminate the regulatory architecture of the human genome, providing novel insight into the consequences of non-coding mutations.

## Methods

**Patients**. A total of 489 patients with idiopathic sporadic NDs and/or multiple CAs with an average age of 10 years (range: newborn to 54 years), comprising 32% females and 68% males, were enrolled in the study (Supplementary Table 1). Inclusion criteria entailed the patient having undergone previous exome sequencing, with no pathogenic findings identified. Many samples tested had also undergone a number of locus-specific tests for common causes of ND–CA, such as Fragile X testing, genomic microarray (Affymetrix 250k SNP array, or Agilent array CGH), and/or WGS. This cohort results from a collaborative effort of multiple centers/groups, namely: 163 trios from The Seaver Autism Center for Research and Treatment at the Icahn School of Medicine at Mount Sinai (USA), 155 trios from the Pediatric Cardiac Genomics Consortium under an approved ancillary study of the PCGC (USA), 94 trios from the Medical Genetics Center Jacinto Magalhaes and the Life and Health Sciences Research Institute (Portugal), and 77 trios from the Nijmegen Medical Center (Netherlands). The main reason for referral was intellectual disability and/or autism spectrum disorder. The majority of patients also presented multiple CAs and/or facial dysmorphisms. A complete list of phenotypic findings is shown in Supplementary Table 1. This study has been conducted in accordance to the rules of the Institutional Review Boards (IRB) of The University of Minho, Portugal, Radboud University Medical Center, The Netherlands, and The Icahn School of Medicine at Mount Sinai, under HS#: 12–00749. Informed consent was obtained from subjects where required. Some samples were

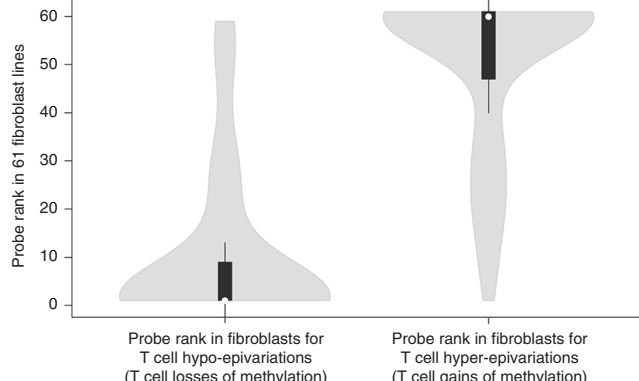

**Fig. 6** Epivariations detected in blood cells are conserved in fibroblasts from the same individual. The presence of outlier methylation changes in T cells is strongly correlated with outlier methylation in fibroblasts from the same individual (Spearman rank correlation 0.75, $p = 1.2 \times 10^{-27}$, Wilcoxon Rank-Sum test). White dots show the median; box limits indicate the 25th and 75th percentiles; whiskers extend 1.5 times the interquartile range from the 25th and 75th percentiles

obtained as residual DNA remaining after clinical testing, and after anonymization were thus not classified as Human Subjects.

Our control cohort resulted from the merger of publicly available datasets taken from the Gene Expression Omnibus (http://www.ncbi.nlm.nih.gov/geo/). In total, we utilized data from 1534 unrelated individuals from the general population without ND–CA who had DNA extracted from peripheral blood and undergone profiling with the Illumina 450k array (GSE36064, GSE40279, GSE42861, GSE53045)[36–39]. GSE36064 focused on a healthy pediatric cohort, GSE40279

analyzed a cohort with a large age range as the goal was to understand epigenetics of human aging, GSE42861 enrolled individuals with rheumatoid arthritis as well as healthy individuals, and GSE53045 included adults who smoked as well as were non-smokers. In total, our control cohort had 60% females and 40% males with an average age of 56 years, range: 1–101 years old. From the samples included in the GEO entries, we excluded a small number of outlier individuals based on principal component analysis (PCA) of autosomal probes. By sequentially comparing each control sample against the remainder using our DMR calling pipeline (see below), we also removed those samples that reported >10 DMRs. A final list of the 1534 controls utilized is available on request.

In order to compare the rate of de novo epivariations in probands to a cohort of healthy individuals, we used dataset GSE56105 from GEO, which comprised 614 healthy individuals from 117 families (mother, father, each with 2–4 children). In order to assess the frequency of epivariations in the general population, we used the dataset GSE55763 from GEO, which comprised 2711 unrelated individuals (40% Type 2 diabetes cases, 60% population controls). Each profile was generated using the DNA extracted from peripheral blood followed by hybridization to the Illumina 450k array[7, 8].

**Methylation array.** Genome-wide DNA methylation profiling was performed using Human Methylation 450k BeadChips (Illumina Inc., San Diego, CA, USA) according to the manufacturer's recommended protocol[40]. Patient samples were processed in six different batches using three different facilities: New York Genome Center (three batches), Genomics Core of the Icahn School of Medicine at Mount Sinai (two batches), and Genetics lab at Northwell Health (one batch). We observed no significant differences in the number of DMRs called per sample, or the rates of secondary validation based on array batch or processing center.

**Quality control and normalization.** Raw data files with $\beta$ values, color, intensity, and detection $p$-values per probe were obtained from the genomics facilities. Quality control steps entailed performing a gender check, a screen for potential regions of homozygous deletion, PCA plots, and density plots of $M$ values. Beta values from autosomes and $\beta$ values from chromosome X were processed and analyzed separately.

We inferred patient gender by calculating both the mean $p$-value of chromosome Y probes, and the mean $\beta$ value of chromosome X probes per sample, and compared these to the patient records. A mismatch between the array-inferred gender and the reported patient gender was detected in four samples: these samples were removed from downstream analysis and the published GEO entry.

As the 450k array utilizes hybridization of DNA to infer a methylation value per probe, regions of homozygous deletion where probes have no target DNA to hybridize often yield very low signal. In such regions of homozygous deletion, reported $\beta$ values are often highly erratic, and thus can easily appear as outlier values when compared to the rest of the population, yielding potential false-positive DMRs in our analysis. We assessed the presence of putative homozygous deletion regions by identifying clusters of probes with failed detection $p$-values. For a high-quality 450k array hybridization, typically <0.1% of autosomal probes yield detection $p$-values >0.01. Thus, clusters of multiple probes in any one sample with failed detection $p$-values ($p$ >0.01) should be extremely rare. The perl script arguments were set to flag 3 kb windows where three or more probes yielded detection $p$ >0.01 in each sample. In total, across the 489 probands, we detected 182 putative regions of homozygous deletion (Supplementary Data 11). By comparing these putative CNVs with publicly available datasets[41] (minor allele frequency, MAF ≥5%), we observed that 81% of clusters of probes with failed detection $p$-values corresponded to common CNV loci or intergenic space, validating this as an appropriate method. Regions identified as putative homozygous deletion in each sample were then excluded from the list of DMR calls.

PCA and density plots of $M$ values were obtained with the lumi and methylumi R packages[42]. We excluded one sample from downstream analysis because it was a clear outlier by PCA. Two additional samples (Proband 488 and Proband 489) were removed from the analysis of chromosome X because they were clear outliers by PCA based on chromosome X data.

After removal of probes with failed detection $p$-values ($p$ > 0.01) in each sample, we performed color correction, background correction, and quantile normalization of $\beta$ values across all probands control samples using lumi and methyllumi. Finally we performed normalization to account for the different data distributions of Infinium type I and type II probes using the BMIQ package[43].

**Identification of candidate DMRs.** Identification of putative DMRs was performed using a custom perl script (available at https://github.com/AndyMSSMLab/Scripts/tree/master). We utilized a sliding window approach to individually compare the methylation profile in each proband against the entire control cohort, detecting regions of robust outlier methylation represented by multiple independent probes with extreme methylation values, including at least one probe with methylation values well outside that observed in any control sample. This was done as single probes can present outlier values due to technical array artifacts, hybridization artifacts such as the presence of underlying sequence variants, or the presence of C>T mutations at the CpG being assayed. Stringent thresholds for calling DMRs were set as follows:

- Hypermethylation: the proband presents, in a 1-kb window, probes that fulfill both of the following criteria:
    (i) At least 3 probes that each have $\beta$ values above the 99.9th percentile of the control distribution for that probe, and are ≥0.15 above the control mean.
    (ii) At least 1 probe with a $\beta$ value ≥0.1 above the maximum observed in controls for that probe.
- Hypomethylation: the proband presents, in a 1-kb window, probes that fulfill both of the following criteria:
    (i) At least 3 probes that each have $\beta$ values below the 0.1th percentile of the control distribution for that probe, and are ≥0.15 below the control mean.
    (ii) At least 1 probe with a $\beta$ value ≥0.1 below the minimum observed in controls for that probe.

All DMRs were manually curated to remove the loci that were deemed false-positive calls. Despite performing probe-level filtering and multiple rounds of normalization of the array data, such measures are imperfect and do not remove all probes that show aberrant signals due to underlying technical or biological effects. We observed both systematic batch effects, and also sporadic false positives in single samples that were filtered, as follows:

(i) Batch effects, i.e., technical differences due to arrays being processed in separate groups, were sometimes observed between cases and controls. Here, it was usually observed that there was either a systematic shift in $\beta$ values reported by one or more probes within a region between arrays processed in different batches. In some cases, the mean of each batch was significantly different, with every sample showing a shift, whereas in other cases the means of the two populations remained similar, but a subset of the samples in one batch showed a gradient of deviations, with the $\beta$ values of multiple cases lying in the extreme tail of the control distribution.

(ii) In some cases, while 3 probes within a 1-kb region were identified as outliers, the outlier probes were not in a contiguous block as would be expected for a true methylation change, and were interspersed with other probes that showed no difference compared to the control population. We interpreted these signals as likely random groupings of individual probes that each yielded outlier beta values for some other reason, e.g., rare underlying variations that influenced probe performance, or poor hybridization performance of individual probes. Indeed, we identified that many such cases were due to regions of homozygous deletion as indicated by clusters of probes with failed array detection $p$-values (Supplementary Data 11).

We observed that 98.6% of the probands presented less than 10 DMRs. Due to a clear increase in the rate of false-positive DMRs as assessed by manual curation, samples with >10 DMRs were excluded (Supplementary Fig. 10). Only the 143 epivariations that were deemed true positive by both researchers were kept for downstream analysis (Supplementary Fig. 2 and Supplementary Data 4). All genomic coordinates are in build GRCh37/hg19. We identified 12 recurrent epivariations (Supplementary Fig. 4). Methylation profiles of 614 healthy individuals from 117 families (GSE56105) were subject to the same analysis, comparing individual methylation profiles against 1534 controls to search for outlier DMRs (Supplementary Data 2). Taking into account the pedigree information, assessment of epivariation inheritance was performed by inspection of data plots of all family members at the DMR locus; 2711 unrelated individuals (GSE55763) were subject to the same pipeline for identification of DMRs (Supplementary Data 3).

**Epivariation calling at imprinted loci.** To identify epivariations affecting imprinted loci, a total of 763 450k array probes mapping to 50 imprinted loci were selected from Monk et al.[44] and Joshi et al.[14]. The mean methylation level was calculated in each proband and control by averaging $\beta$ values for all probes within each imprinted locus. For each proband, methylation changes were considered as epivariations when either the mean methylation level showed a difference greater than 3 standard deviations from the mean of controls, or when the mean $\beta$ value was >0.8 or <0.2. Epivariations identified in imprinted loci are listed in Supplementary Table 1.

**Bisulfite sequencing.** In order to assess the accuracy of our identification of putative epivariations from array data, we performed secondary validation experiments using bisulfite PCR amplicon sequencing. Validation studies were performed in both the proband and parental DNAs (where available) to determine if they were (i) genuine regions of outlier methylation, (ii) inherited from a phenotypically normal parent (and therefore likely unrelated to patient phenotype), or (iii) de novo (and therefore likely pathogenic). Finally, bisulfite sequencing has the additional advantage of being able to determine if the methylation change occurred on one or both alleles, which is important given that the epivariation paradigm predicts that most pathogenic changes will present as mono-allelic gains or losses of methylation.

Samples were processed at the Herbert Irving Comprehensive Cancer Center Epigenetics Medical Center, Columbia University Medical Center. Genomic DNA was bisulfite treated and then subjected to targeted sequencing. Primers were designed using MethPrimer, bisulfite-converted DNA was amplified by PCR, followed by next-generation sequencing (NGS) (Illumina MiSeq). Sample preparation for MiSeq was performed on a Fluidigm AccessArray high-throughput PCR machine with sample bar-codes incorporated in a second round of PCR. Allele-specific methylation was assessed where coverage was >100 reads. Libraries prepared by this method were then subjected to NGS on the Illumina MiSEQ (2 × 150 bp) platform, which scores net methylation in each amplicon based on the ratio of C to T bases at CpG positions. For each amplicon and sample, methylation

percentages (Methylated reads/Total reads) averaged across the covered CpGs (>100×) were provided.

Seventy epivariations were selected for validation with this methodology (Supplementary Data 5). Eleven assays failed to work. Of the remainder, we verified that 33 epivariations were inherited (67% maternal and 33% paternal), 24 were de novo, and 2 were true positives but their inheritance was not assessed (Supplementary Data 4). In some instances, the interpretation of validation experiments was made complex due to highly biased allelic representation, presumably reflecting preferential PCR amplification of one allele (Supplementary Fig. 3).

To confirm the epivariations at imprinted loci, 2 μg of DNA was treated with sodium bisulfite and purified using the EpiTect Bisulfite kit (Qiagen, Germantown MD). Bisulfite PCR for each candidate region was performed on 2 μl of bisulfite-treated DNA using HotStarTaq DNA Polymerase (Qiagen, Germantown MD) and specific primers. After amplification, PCR products were cloned into TOPO TA vector (Invitrogen, Carlsbad, CA) and transformed into chemically competent TOP10 cells (Invitrogen, Carlsbad, CA) for subsequent sequencing using M13R primers (Supplementary Fig. 5).

**Agilent custom designed array**. In order to identify CNVs overlapping or adjacent to epivariations that could underlie the methylation change, we designed a custom ultra-high-resolution CGH array specifically targeting 28 DMRs and ±600 kb of their flanking sequences. At each epivariation locus to be assayed for CNVs, we selected a mean density of 1 probe per ~150 bp ± 100 kb of each DMR, and more sparse coverage (mean density of 1 probe every ~600 bp) extending a further 500 kb upstream and downstream. Normalization probes spaced throughout the autosomes and additional control probes on chromosomes X and Y were also included in the array. The custom array was designed and ordered through Agilent's online portal (https://earray.chem.agilent.com/earray/). DNA samples were processed in the Cytogenetics and Cytogenomics Laboratory of the Icahn School of Medicine at Mount Sinai. The epivariations assayed for CNVs with this method are listed in Supplementary Data 6.

**Targeted sequencing of epivariations**. In order to identify SNVs within DMRs or their flanking sequences that could underlie methylation changes, we performed targeted sequencing of 36 DMRs (Supplementary Data 4), including an additional ±75 kb of sequence from each flank, using a custom sequence capture assay and a HiSeq2500 instrument. Briefly, library preparation entailed shearing DNA using a Bioruptor (Diagenode, Denville NJ) to a mean fragment size of 300 bp, use of the KAPA LTP prep kit and barcoding the DNA with NextFLEX (Bioo Scientific, Austin TX). Capture was performed with a custom oligonucleotide DNA capture kit (Nimblegen, Madison WI). NGS was performed using paired-end 150 bp reads generated with an Illumina HiSeq 2500. Thirty-six samples were multiplexed per sequencing lane, and were processed in the Genomics Core of New York University.

Paired-end reads were mapped against the human reference genome (hg19) using BWA-MEM[45] (https://github.com/lh3/bwa, v0.7.12) with default parameters. Duplicate reads were marked using Samblaster[46] (https://github.com/GregoryFaust/samblaster, v0.1.22). Finally, we used the Genome Analyzer Tool Kit[47] (GATK: https://software.broadinstitute.org/gatk/documentation/article.php?id = 6201, v3.3.0) to perform indel realignment and base quality score recalibration as described in GATK best practices[48]. For manipulating SAM/BAM files and for intermediates steps such as sorting and indexing, we used Sambamba[49] tools (https://github.com/lomereiter/sambamba, v0.5.5). Samblaster was used to generate the file containing discordant reads and split reads, which is required by Lumpy for structural variation calling.

Variant discovery was performed using GATK's n+1 joint genotyping protocol (https://software.broadinstitute.org/gatk/documentation/article.php?id = 3893). The protocol involves multiple steps. Initially, the gVCF file was created individually for each sample using the haplotype caller utility. Next, joint genotyping was performed on all samples together and a single vcf file generated. In all steps, we restricted genotyping to the targeted loci. The resulting variants calls were annotated with CADD[50] scores (http://cadd.gs.washington.edu/, v1.3), MAF from the 1000 Genomes Project (phase 3) using Annovar[51] (http://annovar.openbioinformatics.org/en/latest/, v2016Feb01). Rare variants (1000 Genomes MAF < 1%) were further annotated with transcription factor (TF)-binding sites predicted by the CENTIPEDE algorithm[52], and DNAseI hypersensitivity sites from the ENCODE project[53] (Supplementary Data 7). Where parental genotypes were available, concordance of inheritance between an epivariation and SNP were used to filter candidates. In order to assess if these rare SNVs were disrupting TF-binding sites (TFBS), we used the UCSC Genome Browser track "Transcription Factor ChIP-seq (161 factors) from ENCODE with Factorbook Motifs"[54] (http://hgdownload.soe.ucsc.edu/goldenPath/hg19/database/factorbookMotifPos.txt.gz, 16 March 2014 release), which included canonical TFBS motifs for 129 TFs.

Putative duplication and deletion structural variants (SV) were called jointly using Lumpy[55] (https://github.com/arq5x/lumpy-sv, v0.2.12). Lumpy identifies SV breakpoints, which were then genotyped using a Bayesian genotyper, SVTyper[56] (https://github.com/hall-lab/svtyper, v0.0.4). SVs with allele balance (AB) <0.1, read support <5, and genotype quality <30 were removed. Also all sites genotyped as missing (./.) or homozygous ref (0/0) were removed. Further, SVs present in >10% of samples were removed. The resulting set of filtered SVs were visually validated using Integrative Genomics Viewer (IGV)[57] (Fig. 3).

**Epivariations in B cells, T cells, and fibroblasts**. We obtained raw DNA methylation data generated using the Illumina 450k Human Methylation BeadChip from the Gencord cohort from the EMBL-EBI European Genome–Phenome Archive (https://www.ebi.ac.uk/ega/) under accession number EGAS00001000446, representing 107 fibroblast cultures, 66 T-cell cultures, and 111 immortalized B-cell cultures derived from a cohort of newborns[21]. As described above, we performed *lumi* and BMIQ normalization. Given that we lacked a large set of control samples of matched cell type for comparison, DMRs in each sample were identified as outliers relative to the rest of the population, using a 1-kb sliding window. In each window, we required at least 3 probes with β value ≥0.15 the maximum, or ≥0.15 below the minimum, of that observed in all other individuals. Based on the β values of each probe located within the 1 kb DMR loci, we calculated the population rank of each individual carrying an epivariation defined in T cells and fibroblast methylation profiles (Supplementary Data 9).

**Gene expression studies in 90 lymphoblastoid cell lines**. We used normalized methylation data for a filtered set of 443,498 probes in 90 samples analyzed as part of the 1000 Genomes Project for which both variant calls and RNAseq data were also available (GEO GSE39672)[19, 20]. DMRs were called using an outlier approach using a 1-kb sliding window, with at least 3 probes with β value ≥0.15 the maximum, or ≥0.15 below the minimum, observed in the other 89 individuals. RNAseq (http://www.ebi.ac.uk/arrayexpress/files/E-GEUV-1/analysis_results/) and SNV data[18] were obtained and used to measure total gene expression, and allelic expression levels based on heterozygous transcribed SNVs. For the latter, at each transcribed SNV position with at least 7 overlapping reads, we made counts of the number of reads containing reference and alternate alleles. In total, there were 300,111 sites within Refseq gene annotations with at least one individual carrying a heterozygous SNV with read depth ≥7. We linked each DMR with associated genes based on physical overlap with gene promoter regions, defined as ±2 kb from the annotated transcription start site (TSS) (Supplementary Data 10).

**Analysis of SNVs around DMRs in 90 lymphoblastoid cell lines**. We downloaded SNV data for 90 controls from the 1000 Genomes Project (ftp://ftp.1000genomes.ebi.ac.uk/vol1/ftp/phase3/integrated_sv_map/ALL.wgs.integrated_sv_map_v2.20130502.svs.genotypes.vcf.gz.). Using the coordinates of each DMR called in these 90 samples, we extracted SNVs located within ±5 kb of each DMR, yielding a total of 20,398 DMR-SNV pairs. Variants were then filtered to retain only those with MAF < 0.1% in the total 1000 Genomes population (Supplementary Data 8), and annotated for the overlapping TFBS based on ENCODE/Factorbook data, as described above. TFBS enrichment analysis utilized the 89 individuals without a DMR as background for each test.

**Conservation of epivariations across multiple tissues**. We downloaded from GEO (GSE48472) the methylation profiles generated using the Illumina 450k array from five deceased individuals, each profiled in six different tissues (peripheral blood, liver, skeletal muscle, pancreas, omental fat, and spleen)[24]. Data for each tissue were filtered and normalized separately, following the same approach as used for patient methylation profiles, as described above (see "Methylation array"). For the peripheral blood data, we then quantile normalized β values with those for the 1534 controls. DMRs were then called in each of the six blood samples using the same approach as described above (see "Methylation array"). For each DMR observed in blood, we generated plots of these loci and manually curated for concordance in the other available tissues (Supplementary Fig. 9).

**Code availability**. Computer code used in this study is available on GitHub: https://github.com/AndyMSSMLab/Scripts/tree/master

**Data availability**. The methylation array data used in this publication have been deposited in Gene Expression Omnibus (GEO) under accession GSE89353 (https://www.ncbi.nlm.nih.gov/geo/query/acc.cgi?acc=GSE89353).

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

## Acknowledgements

The authors are grateful to the patients and families who participated in this study and to the collaborators who supported patient recruitment. This work was supported by NIH grant HG006696 and research grant 6-FY13-92 from the March of Dimes to A.J.S., grant HL098123 to B.D.G. and A.J.S., Gulbenkian Programme for Advanced Medical Education and the Portuguese Foundation for Science and Technology (SFRH/BDINT/51549/2011, PIC/IC/83026/2007, PIC/IC/83013/2007, SFRH/BD/90167/2012, Portugal) to P.M., F.L., and M.B., by the Northern Portugal Regional Operational Programme (NORTE 2020), under the Portugal 2020 Partnership Agreement, through the European Regional Development Fund (FEDER) (NORTE-01-0145-FEDER-000013) to P.M., a Beatriu de Pinos Postdoctoral Fellowship to R.S.J. (2011BP-A00515), and a Seaver Foundation fellowship to S.D.R. The views expressed are those of the authors and do not necessarily reflect those of the National Heart, Lung, and Blood Institute or the National Institutes of Health. Research reported in this paper was supported by the Office of Research Infrastructure of the National Institutes of Health under award number S10OD018522. This work was supported in part through the computational resources and staff expertise provided by Scientific Computing at the Icahn School of Medicine at Mount Sinai.

## Author contributions

M.B., R.S.J., P.G., H.G.B., J.D.B, B.D.G., and A.J.S. were leading contributors to the design and analysis of this study; M.B., T.K., D.E.G., G.S., P.M., H.G.B, J.D.B., and B.D.G. contributed with samples of probands and relatives; M.B., D.E.G., S.D.R., J.R., F.L., P.M., L.V., T.K., and G.S. contributed with patient clinical/genetic information; P.G., N.P., B.J., C.T.W., and K.C. wrote and performed bioinformatic analysis; A.M.T. analyzed and validated the methylation profiles of imprinted loci; W.G. performed library preparation and capture for targeted sequencing; C.T. contributed for Agilent custom designed aCGH; H.M. and L.E. processed the Agilent custom designed aCGH; M.B. and A.J.S. wrote the manuscript, all authors commented on it.

## Additional information

**Competing interests:** The authors declare no competing interests.

