## [Peer Review File · Nature Communications]

Reviewer #1 (Remarks to the Author):

Barbosa et al present a method for detecting methylation disturbance in 450k/EPIC array data, which is a potential proxy for altered transcriptional activity, offering a means of interpreting noncoding genomic variation and diagnostic uplift for rare disorders. They present validation of array results and detection of relevant genomic changes.

I believe this is potentially as valuable an approach as CNV analysis was when it emerged some years ago – and presents as many challenges for empirical determination of pathogenic versus polymorphic variation. I believe it is fundamentally very worthy of publication; but I have some comments where I'd like the authors' feedback.

1. I wonder why the authors have submitted to Nature Communications. NC has high esteem, wide readership and a broad range of articles; but it seems to me better suited to basic science than to – as I believe this paper ought to be – a first presentation of a new clinical/molecular diagnostic method. Perhaps it is because the authors have not invested heavily in the clinical characteristics of their cohort or exploring the interplay between molecular and clinical interpretation. This feels like a group of gifted scientists not engaging with the clinical questions related to their work.

It would not have been hard for the authors to analyse DNA methylation in cases with known coding or structural variations leading to clinical disease, to directly demonstrate that variation in established locations with clear clinical consequences is indeed associated with methylation variation.

2. Questions on the epigenomic analysis.

The authors have followed standard QC and pre-processing protocols for the epigenomic analysis. In general the 'DMR' method is quite statistically robust based on the criteria presented. However, my main concern here is that there is no robust justification for the threshold set; it appears empirical and not based on robust control data. For example: no reason is given for setting the methylation change threshold at 0.15 or 10% increase/decrease in methylation level compared to control samples.

The stringent thresholds for calling 'DMRs' in the supplementary methods (Lines 82-107) are quite complicated and compact, and not readily understandable for this reviewer. A breakdown of the criteria, fully explained, would be more helpful. There appears to be an error in the criteria, for hypo/hyper methylation (supplement lines 93-98), in that one requires methylation value ≥ 1 probe \geq maximum/minimum $\pm 0.1 \beta$ value observed in controls, and the other ≥ 1 probe \leq maximum/minimum $\pm 0.1 \beta$ value; this isn't reassuring. Is it exactly one probe or minimum 1 probe – and why? Using exactly one probe does not make any sense to me.

Can the authors clarify their statistical / biological / technical rationales for excluding samples having >10 DMRs (Line 99); this also appears based on empirical observation rather than statistical stringency.

At Line 88, determining 'extreme methylation values well outside that in any control sample' is not clear. What do the authors mean by 'any control sample'? Single controls, or groups of control samples?

Though the authors mentioned no significant differences in the number of DMRs called per sample, or the rates of secondary validation based on array batch or processing center, the basis of the observation is not clear. Authors should have done batch correction prior to the DMR analysis.

It remains unclear to me, fundamentally, how batch correction was performed to enable comparison of single sample methylation data with groups of controls arrayed at different times. If the authors have a robust method for liberating methylation arrays from the challenge of batch effects, they should state it very clearly with diagrams and workflows because it is quite useful.

Was there correction for age and tissue origin?

3. I have a problem with some of the authors' nomenclature and terminology.

(a) The authors refer to epimutations (starting lines 57, 60,61,62). In genomics we have been getting very used to avoiding the term 'mutation', because a sequence variant's pathogenicity varies according to its genetic and biological context. It is irresponsible to use the term 'epimutation' unequivocally when the pathogenicity of these changes is not demonstrated and 58% (line 141) are shared by – we guess – unaffected parents. The authors could use 'epivariation' or an alternative.

(b) the authors refer to 'DMRs' (line 108) as strong outliers from control methylation values. As mentioned above, their definition of 'strong outlier' needs further justification. DMRs are recognised from the field of genomic imprinting as specific genomic sequences subject en bloc to parent of origin-specific divergence in methylation. Co-opting the term DMR for undefined idiogenic regions showing DNA methylation variation – particularly when imprinting is also relevant in some cases – is potentially confusing. Perhaps DMV (DNA methylation variant, reflecting CNV) or some other term may be invented.

(c) "a 95% true positive rate" (line 94). A true technical positive, I guess – confirmation of the methylation level by an orthogonal test?

(d) "some cases of ND/CA ... harbor epigenetic aberrations that lead to a dysregulation of normal genome function" (lines 91-92). These epigenetic changes probably biomark dysregulated genome function. The authors are over-claiming here.

(e) "epimutations represent large methylation changes specifically on one allele, with most showing two clusters of largely methylated and unmethylated reads". Some of the samples designated 'true positive' in Supp Table 5 show <10% change in methylation; this does not warrant use of the blanket term 'large' (specially twice in one sentence).

(f) "We also observed that many hypermethylated epimutations at promoters are associated with complete silencing of one allele (Extended Data Fig. 7), and, thus, have an impact comparable to that of loss-of-function coding mutations" It is true that Extended data fig 7 shows an association between differently-methylated alleles and allele-specific expression in one individual. This does not

demonstrate a causative relationship. Experimental demonstration is required for the authors to make the claim they do. In my opinion, as a result, the closing statement “Our study shows ... that epimutations ... often exert strong functional effects on gene expression” is not justified – in particular the qualifier ‘often’.

4. Lines 138-143 are not clear to me. What I am taking from these words is that apparently healthy individuals and individuals with ND/CA both harbour variations in DNA methylation, with a degree of enrichment on a purely numerical basis in the individuals with ND/CA, and an interesting question about whether this is related to their clinical condition. Can the authors clarify their claim here?

5. I am having some problems with the authors’ view on the clinical significance of methylation changes in their case cohort.

As mentioned above, in 58% of samples where parents were available, epivariations were shown to be present in apparently healthy parents. Perhaps the discussion can explore the likely prevalence of methylation variants in the population, as has been done for CNVs.

Was any evidence sought or found that shared clinical features (e.g. the heart disease mentioned in a significant number of the case cohort) were associated with shared epigenetic changes?

“three epimutations encompassed the promoters of genes with prior known disease associations and/or hypermethylated triplet repeat expansions (MEG3, FMR1, FRA10AC1), validating our method for detecting pathogenic epimutations” (lines 155-158). This feels like a moment of linguistic sleight of hand, suggesting clinical validation of pathogenicity. As it happens, one of the individuals with a MEG3 change does have clinical features in common with an imprinting disorder. But since the authors don’t in any way address specific clinical cases, it is important that they should not make claims that might be taken to indicate clinical validity.

“This represents a significant enrichment for rare SNVs disrupting CTCF binding sites ... when compared to the same regions in controls” (lines 176-178). I am not clear from the text that the authors sequenced their control cohort in the same way they sequenced their cases. Can this be made explicit, so this significant enrichment can be justifiably stated?

“Our study shows for the first time that epimutations are ... implicated in developmental disorders” (lines 247-248). This claim is not justified.

6. I am not finding the data related to the assertion in lines 234-245. It feels rather unprepared, unsubstantiated and isolated tucked in just before the discussion section, and doesn’t seem to fit in with the rest of the data; perhaps it could be omitted.

Reviewer #2 (Remarks to the Author):

The paper by Barbosa et al is very exciting. Although set in the context of NAs and NDs the paper is relevant to a vast amount of research that is being conducted to understand the genetic aetiology of human disease/disorders that is almost universally falling short of its ambition. This paper explores an alternative with considerable and optimistic outcomes. The resources and context of the study are excellent.

Some care needs to be given to language in this manuscript.

Epimutation

The term “epimutation” is strictly defined as a heritable change in gene activity that is not associated with a DNA mutation but rather with gain or loss of DNA methylation or other heritable modification of chromatin. Changes in gene expression through altered DNA methylation or histone modifications induced from cis- or trans-acting genetic factors known as methylation Quantitative Trait Loci, (mQTL) are therefore not epimutations in this strict sense. Please modify the text accordingly.

In some instances the manuscript is difficult to follow without extensive reference to the Supplementary material. Could the authors provide further information around the following aspects to improve the flow of the presentation without extending the text unreasonably? Eg

After stringent quality control... line 109

Each sample was screened for epimutations...line 107

Secondary validation an assessment....line 112

The brief methods state that an “epimutation” (see above) is defined as strong outlier differentially methylated regions absent in the 1,534 controls – how were the outlying control (as described and shown in Figure 1) managed in the analysis?

Figure 2. I think this would be better presented as total n rather than percentage. Please also check the numbers quoted – are de novo “epimutations” n = 49 (legend) or n = 24 (Figure)? Are the number of de novo “epimutations” in controls 6 (figure) or 40 (legend)?

Figure 3. There are two green lines in this figure – could the colour coding be changed?

What does “often strongly” mean at line 203?

I do not think that CTCF is defined anywhere in the main text.

Line 68 and 94: What types of microarrays are the authors referring to and what are they measuring?

Thank you for sharing this with me, excellent paper.

Reviewer #3 (Remarks to the Author):

Barbosa et al. perform a deep and comprehensive analysis to identify DMR patterns in ND/CA probands and characterize them in a most detailed fashion. We find this study covers the subject very extensively, and reveals possible correlation between epigenetic patterns, gene expression and the ND/CA. We have raised several issues that should be considered and/or explained before this manuscript is considered for publication. Below are more major comments, followed by minor ones:

1. CTCF participates in methylation activity, thus the canonical splice site SNVs and CTCF's regulation function can definitely explain the DMRs in that vicinity. A disrupted CTCF protein could also correlate with methylation dysregulation throughout the genome and by that explain the enrichment of DMRs of those specific cases. Can this bias the overall DMR enrichment in cases (observed in line 142?)
2. Are the 70 samples the only ones that were validated using bi-sulfite? (line 113)? Why were those samples selected and how was the selection process performed? Also, if we understand correctly, 24/70 epimutations were verified as de-novo events (34%, extended data line 143-146). The authors point this as 42%(??). Furthermore, this seems like a very high proportion of new to inherited events. How do the authors explain this number? What is the proportion to be expected

from literature? On the same subject, 117 control families were used to assess de novo rate in controls. What method was used to call the de novo epimutations in the controls?

3. How did the authors test for significance of enrichment in cases vs. controls? How did they correct for multiple testing deriving from multiple regions in the microarray? This is also relevant for the CTCF enrichment in line 177 since 50 other DMRs were also tested. Regarding the SNVs in CTCF: what was the number of control variants in this location? What is the population allele frequency in general population? (ExAC/gnomAD?)

4. How did the authors correct for gender? cases have 68% males and controls 40%.

5. The average age for cases – 10 years, controls is 56 years old. Methylation signatures are being accumulated with age, thus the test of young to old makes less sense.

6. GSE55763 is comprised of 2,664 controls and 36 technical replicates, but authors claim they used 2,711 controls from that dataset. Also, what is the nature of these controls? Do they belong to a specific age/phenotypic group that might affect their methylation patterns?

7. What are the criteria to decide on an epimutation (line 110)? DMRs were rated visually by two researchers (Extended data line 100). What were the parameters by which each decided on a true positive DMR? What was considered as false positive?

8. While nicely depicted, the effect of hypo/hyper methylation of promotor regions (lines 200-208) has long been known and discussed. What is the benefit of this analysis to what was previously known? The authors should recognize previous studies describing this phenomenon. Also, the correlation between methylation patterns between tissues (lines 223-232) has also been deeply described previously. For example, see: Lohk et al, 2014, Genome Biology, or the more recent Guo et al, Nat Genetics, 2017. What is the benefit of this study compared to what was previously described?

9. Also regarding gene expression changes with regards to methylation patterns: we could not find the relevant information regarding the numbers of genes that are hypo/hyper methylated and their expression differences.

10. Figure 1 - There's no legend and no text to relate to the various panels.

11. Figure 1 panel A – if parents are 50% methylated and child is 20%, why is that considered De-novo?

12. No explanation of the regions that were chosen for panels A-C in the text

13. MOV10L1 (hg19: chr22:50528178-50528751): this region is not just the promoter. it covers the 5' UTR as well as first exon and part of intron

14. We are uncomfortable with the use of the CADD score for ##. Several recent studies show CADD to have large amounts of false positive variants (low specificity) in non-coding regions

(Gelfman et al. 2017, Mather et al. 2016, van der Velde et al. 2017, Shihab et al. 2015, etc). Some of the other scores present significantly better tp/fp rates in direct comparison with CADD for various non-coding annotations.

15. What is the difference in calling methods for epimutations between cases and the different sets used for controls?
16. What is the rate of recurring epimutations in controls? How many controls share the same epimutations as were shared between cases (lines 156-157)
17. How were the specific 50 DMRs chosen for targeted sequencing? (line 170)
18. Extended Data Figure 4: where are the Bisulfite sequencing validations of the proband for HM13. Also, legend read both empty and full circles as “Methylated CpG”
19. Also, there is no chance to read the text in Extended figure 9 to know which graph is which tissue.
20. Extended Data Figure 5: looks very messy, should this be divided into panel A+B in one figure, C and D as separate figures?
21. Extended Data Figure 6: The result is written in the legend of the figure

Reviewer #1 (Remarks to the Author):

Barbosa et al present a method for detecting methylation disturbance in 450k/EPIC array data, which is a potential proxy for altered transcriptional activity, offering a means of interpreting noncoding genomic variation and diagnostic uplift for rare disorders. They present validation of array results and detection of relevant genomic changes. I believe this is potentially as valuable an approach as CNV analysis was when it emerged some years ago – and presents as many challenges for empirical determination of pathogenic versus polymorphic variation. I believe it is fundamentally very worthy of publication; but I have some comments where I'd like the authors' feedback.

1. I wonder why the authors have submitted to Nature Communications. NC has high esteem, wide readership and a broad range of articles; but it seems to me better suited to basic science than to – as I believe this paper ought to be – a first presentation of a new clinical/molecular diagnostic method. Perhaps it is because the authors have not invested heavily in the clinical characteristics of their cohort or exploring the interplay between molecular and clinical interpretation. This feels like a group of gifted scientists not engaging with the clinical questions related to their work. It would not have been hard for the authors to analyse DNA methylation in cases with known coding or structural variations leading to clinical disease, to directly demonstrate that variation in established locations with clear clinical consequences is indeed associated with methylation variation.

Thank you for the positive comments on our work, we agree that this is a highly significant piece of research, which in our opinion is of high impact in the field. In fact, prior to submitting to *Nature Communications* we had sent the manuscript to *Nature Genetics*. In rejecting it, the editors suggested *Nature Communications* as an excellent alternative. We obviously took that advice and, based on the initial reviews, feel encouraged that that this is an appropriate journal for this manuscript. Some of the manuscript formatting was a result of those earlier submissions and their associated limitations. We have now added additional text to the manuscript to more adequately describe some of the points raised during review, and also moved two figures from Supplementary Information to the main manuscript to highlight these results better.

With regard to the specific comment of analyzing DNA methylation patterns in cases of structural variations or coding mutations, work towards this has already been done. My own lab had looked at multiple cases of apparently balanced translocations associated with clinical phenotypes, but we were unable to detect any epigenetic modifications associated with these translocation events (unpublished data). We have also studied several X;autosome translocations and demonstrated a clear epigenetic effect of the spreading of X-inactivation into autosomal DNA (PubMed ID: 24186870). Other groups have reported specific DNA methylation profiles in a subset of patients with coding mutations of genes involved in epigenetic regulation and chromatin modification, such as those that underlie Sotos, CHARGE and Kabuki syndromes (PubMed ID: 26690673 and 28475860). However, all of these have a distinct etiology compared to the phenomenon we report in the current manuscript. Rather than analyzing patients with known structural variations or single-gene defects, here we were specifically testing the hypothesis that patients in whom prior genetic testing had failed to identify a causative defect might harbor epigenetic defects.

2. Questions on the epigenomic analysis.

The authors have followed standard QC and pre-processing protocols for the epigenomic

analysis. In general the ‘DMR’ method is quite statistically robust based on the criteria presented. However, my main concern here is that there is no robust justification for the threshold set; it appears empirical and not based on robust control data. For example: no reason is given for setting the methylation change threshold at 0.15 or 10% increase/decrease in methylation level compared to control samples.

The stringent thresholds for calling ‘DMRs’ in the supplementary methods (Lines 82-107) are quite complicated and compact, and not readily understandable for this reviewer. A breakdown of the criteria, fully explained, would be more helpful. There appears to be an error in the criteria, for hypo/hyper methylation (supplement lines 93-98), in that one requires methylation value ≥ 1 probe \geq maximum/minimum $\pm 0.1 \beta$ value observed in controls, and the other ≥ 1 probe \leq maximum/minimum $\pm 0.1 \beta$ value; this isn’t reassuring. Is it exactly one probe or minimum 1 probe – and why? Using exactly one probe does not make any sense to me.

The criteria we utilized in the manuscript for identifying DMRs were developed over a period of >1 year, and went through multiple iterations and refinements to ensure that we were identifying large and rare changes in methylation as robustly as possible. We initially set criteria based on prior work we had done studying loss of imprinting defects caused by uniparental disomy (PubMed ID: 27569549). In that study, the availability of data for dozens of loci that showed allelic changes in methylation levels provided us clear examples of the type of events we were looking for. Based on this we initially set criteria that were then further refined as we processed increasing numbers of samples, with an aim of minimizing false positives. That we have achieved this aim is strongly demonstrated by our validation rate of 95% by bisulfite PCR/sequencing. In this sense, the thresholds were set empirically, but do function well to maximize the number of true positives while maintaining a low false positive rate. Once we had gained reasonable numbers of loci validated by bisulfite PCR, we then further refined our thresholds, eventually settling on the criteria listed in our manuscript, as we found that if we further increased our stringencies we were no longer able to detect loci that we had already validated as representing genuine differences. Particularly if these methods are to be used in a clinical testing environment, future work will be needed to optimize the tradeoffs between sensitivity and specificity.

We apologize for not explaining our DMR calling thresholds in the manuscript clearly. We have therefore modified the text in the Supplemental Methods to make this clearer, so that it now reads as follows:

“Stringent thresholds for calling DMRs were set as follows:

- Hypermethylation: the proband presents, in a 1-kb window, probes that fulfill both of the following criteria:
 - (i) At least 3 probes that each have β values above the 99.9th percentile of the control distribution for that probe, and are ≥ 0.15 above the control mean,
 - (ii) At least 1 probe with a β value ≥ 0.1 above maximum observed in controls for that probe.
- Hypomethylation: the proband presents, in a 1-kb window, probes that fulfill both of the following criteria:
 - (i) At least 3 probes that each have β values below the 0.1th percentile of the control distribution, for that probe, and are ≥ 0.15 below the control mean for that probe,
 - (ii) At least 1 probe with a β value ≥ 0.1 below the minimum observed in controls for that probe.”

We thank the reviewer for pointing out our error regarding the way we expressed that probes in hypomethylated DMRs. This should have been written “ ≥ 0.15 below the control mean” – we

apologize for the confusion this caused. Also we required at least one probe that shows a difference that is ≥ 0.1 beyond the most extreme value observed in controls, not exactly one.

Can the authors clarify their statistical / biological / technical rationales for excluding samples having >10 DMRs (Line 99); this also appears based on empirical observation rather than statistical stringency.

We decided to exclude samples that had >10 DMRs due to low specificity. After our initial algorithmic DMR calling, all calls were further curated by visual inspection of plots. We found a clear relationship between the number of original DMRs called by the algorithm and the fraction that were discarded as false positives. For example, for samples with 1-9 DMRs, 83% of the original calls were true positives. However, for samples with ≥ 10 DMRs, 89% of the calls were false positives. This dramatic reduction in specificity in samples with >10 DMRs strongly suggested that these samples were inherently problematic, probably for technical reasons. Therefore, in order to be conservative and maintain a high specificity, we excluded these noisy samples from our study. We have added the following statement to the Supplemental Methods, such that it now reads as follows:

“Due to a clear increase in the rate of false-positive DMRs as assessed by manual curation, samples with >10 DMRs were excluded (Extended Data Fig. 10).”

At Line 88, determining ‘extreme methylation values well outside that in any control sample’ is not clear. What do the authors mean by ‘any control sample’? Single controls, or groups of control samples?

We meant that a DMR was defined as a region showing methylation values that were not observed in any of the 1,536 controls. At each probe position in the genome, we calculated the minimum and maximum value observed in the 1,536 controls, and required that a DMR called in a patient have at least one probe with a beta value at least 0.1 greater than the maximum of controls (for hypermethylated DMRs), or at least 0.1 less than the minimum value observed in controls (for hypomethylated DMRs). In addition, DMRs required at least two other probes to be within the 0.1% most extreme tails of the distribution observed in controls. In addition, all three probes were required to have a difference ≥ 0.15 from the control mean.

Though the authors mentioned no significant differences in the number of DMRs called per sample, or the rates of secondary validation based on array batch or processing center, the basis of the observation is not clear. Authors should have done batch correction prior to the DMR analysis. It remains unclear to me, fundamentally, how batch correction was performed to enable comparison of single sample methylation data with groups of controls arrayed at different times. If the authors have a robust method for liberating methylation arrays from the challenge of batch effects, they should state it very clearly with diagrams and workflows because it is quite useful. Was there correction for age and tissue origin?

Our data pre-processing included all samples (cases and controls together), and performed quantile normalization of intensities in each color channel using *lumi*, followed by BMIQ which applies further quantile normalization of beta values based on probe type. These two steps together are standard methods that act to normalize together Illumina methylation array data produced from different batches, although as with any normalization technique, they will reduce rather than completely eliminate technical batch effects. We do not claim to have a method that removes batch effects, merely we observe that our particular approach for identifying DMRs

seems to be relatively robust to batch effects when one uses a sufficiently large control population. This observation was made on the basis that although PCA plots of our case data versus control data could clearly distinguish each batch as separate in PCA space, we did not observe any distinct increase in the rate of false positive DMRs that we identified in any single batch of data. Below we include some PCA plots of raw array data to show that each batch of case and control data was distinct, but even where we had data from cases that was entirely separable from controls using PCA, we did not observe any excess of false positive DMR calls.

Response Figure 1. Example PCA plots showing the distribution of data from different batches of arrays run on cases and controls. (left) This plot shows data for a batch of cases that we generated (red) and control samples downloaded from GEO (blue). Although cases and controls are clearly separable in PCA space, we did not observe any obvious excess of false positive calls in this batch of cases compared to others we used that seemed to be more similar in PCA space to the set of controls. **(right)** This plot shows data for the multiple different control cohorts that we downloaded from GEO, together with a batch of cases (labeled with black “R”) using data we generated. Each array batch is colored separately, and the different batches of data can be clearly observed as occupying distinct PCA space. For this batch of cases they corresponded in PCA space fairly well with several of the controls batches, but we still observed a comparable rate of DMRs called in this batch as all others.

No correction was applied to our data for age or tissue. In other contexts, *e.g.*, EWAS analysis, comparing case data to control data that were each produced in different batches would almost certainly simply identify technical effects in the data due to differences in array processing. Similarly, without appropriately controlling or correcting for age, cell composition and gender differences that exist in cases versus controls, many effects seen **on a population level** would likely be due to these inherent population stratifications. However, unlike an EWAS, the design of our study was fundamentally different: here we merely ask the question “where do we see a methylation pattern in the genome of a single individual that is more extreme than that observed in any of our 1,536 controls”. As our control population is much larger than our case population and contains individuals of both genders with the full spectrum of ages from newborn to >90 years old, there is no need to perform adjustments for age or gender. Similarly, as our control population is composed of a large number of individuals, it is reasonable to presume that these control individuals will have blood counts that will therefore span the full spectrum of blood compositions seen in the general population. It is also known that most methylation differences due to age, gender and changes in cellular composition of blood tend to be fairly small in magnitude. As our analysis algorithm requires that a case individual shows a methylation profile that has multiple probes showing more extreme beta values than those observed in any of the

controls, epigenetic variations due to age, gender and blood counts will likely have little impact on our results, except perhaps to cause some false negatives, although this is difficult to quantify. Finally, if any systematic bias such as age or gender were significantly affecting our results, we would expect to see many age and gender related loci identified as DMRs across multiple samples – this was not the case. Overall, given that our independent validations with bisulfite sequencing showed a true positive rate of 95%, this strongly argues that our data normalization approach and criteria for identifying DMRs is robust to batch effects.

Furthermore, while there are methods to estimate age and cellular fractions from methylation data, these cannot be used to adjust or correct the beta values in a sample. Instead they merely provide co-variables that one can include in e.g. regression analysis of a population, which would not be useful in our scenario where we are directly comparing a single sample against a control cohort using beta values.

3. I have a problem with some of the authors' nomenclature and terminology.

(a) The authors refer to epimutations (starting lines 57, 60,61,62). In genomics we have been getting very used to avoiding the term 'mutation', because a sequence variant's pathogenicity varies according to its genetic and biological context. It is irresponsible to use the term 'epimutation' unequivocally when the pathogenicity of these changes is not demonstrated and 58% (line 141) are shared by – we guess – unaffected parents. The authors could use 'epivariation' or an alternative.

We have changed our use of this term and instead use the term “epivariation” throughout the manuscript, including revising the manuscript title.

(b) the authors refer to 'DMRs' (line 108) as strong outliers from control methylation values. As mentioned above, their definition of 'strong outlier' needs further justification. DMRs are recognised from the field of genomic imprinting as specific genomic sequences subject en bloc to parent of origin-specific divergence in methylation. Co-opting the term DMR for undefined idiogenic regions showing DNA methylation variation – particularly when imprinting is also relevant in some cases – is potentially confusing. Perhaps DMV (DNA methylation variant, reflecting CNV) or some other term may be invented.

We have changed the term “strong outlier” to “rare outlier” as, by definition as being absent from >1,500 controls, these are rare events.

While we agree with the reviewer that the term “DMR” is frequently used in association with sites that show parent-specific methylation (*i.e.*, imprinting), this is in no way its sole usage in the literature. In fact, differentially methylated region (or DMR) is a term that is more frequently used in much broader terms in the genetics literature in association with any locus that shows a methylation difference between alleles, tissues or groups (e.g., populations of cases and controls). For example, it is frequently used in the cancer literature, in epigenome-wide association studies, or in studies of epigenetic alterations of disease states (see <https://www.ncbi.nlm.nih.gov/pubmed?term=Differentially+methylated+region>)

To illustrate this, albeit in a rather crude way, a PubMed search for the term “Differentially methylated region” yields 946 hits, while a search for the term “Differentially methylated region AND imprinting” yields 413 hits. Thus, DMR is commonly used to refer to regions of differential methylation that are not imprinted, and as such we feel that its use in this manuscript to refer to regions that show different methylation status is an appropriate one.

(c) “a 95% true positive rate” (line 94). A true technical positive, I guess – confirmation of

the methylation level by an orthogonal test?

We have edited this sentence such that it now reads:

“Using PCR/bisulfite sequencing, we performed orthogonal confirmation for 70 epivariations (Supplementary Table 5), yielding a 95% true positive rate.”

(d) “some cases of ND/CA ... harbor epigenetic aberrations that lead to a dysregulation of normal genome function” (lines 91-92). These epigenetic changes probably biomark dysregulated genome function. The authors are over-claiming here.

At this point in the manuscript we are simply stating our initial hypothesis for the study, not making any claim. , To clarify further, we have edited the phrase so that it now reads as follows:

“We hypothesized that some cases of ND/CA that remain refractory to conventional sequence-based analysis harbor epigenetic aberrations that are associated with dysregulation of normal genome function.”

(e) “epimutations represent large methylation changes specifically on one allele, with most showing two clusters of largely methylated and unmethylated reads”. Some of the samples designated ‘true positive’ in Supp Table 5 show <10% change in methylation; this does not warrant use of the blanket term ‘large’ (specially twice in one sentence).

Below we show a bean plot of the mean difference in methylation per DMR, where we averaged all CpGs tested by bisulfite sequencing in the proband vs controls within the DMRs scored as true positives, as listed in Supplementary Table 5. It can be seen that almost all DMRs report mean methylation changes that are >0.15, and median change is >0.3.

Response Figure 2. Bean plot showing mean difference in methylation for all DMRs assessed by bisulfite sequencing.

There are two loci shown above and in Supplementary Table 5 that from the bisulfite sequencing report a mean difference in proband vs control mean of just 2.5% and 6.8%, which

we agree is small. However, in both of these cases we feel that the average difference of CpGs tested by bisulfite sequencing does not present an accurate summary of the changes actually observed at these two loci. This may be because in some cases, due to the limits of bisulfite PCR primer design, we were not able to target the amplicon to the core DMR identified by array, but instead it was located at the edge of the DMR, and as a result these PCR assays were only detecting small differences. As a result, for these two loci the numbers listed in the final two columns are not representative of the DMRs we originally observed. Below we show plots of the original 450k array data for these two DMRs. In both cases it can be seen that there are multiple CpGs that all show large changes in the proband (>0.3 change in beta value) when compared to the mean of the control cohort. Based on the array data, probes in the DMR of *FZD6* showed a mean difference in proband vs control of 0.34, while for *MOV10L1* DMR probes showed a mean difference of 0.14. In this latter example, the mean difference for DMR CpGs tested by array is again diluted by the fact that there are several probes that show small differences in proband vs controls, and in fact there are several probes that each show differences >0.3. As such, we feel that our original statement in the manuscript that “all epivariations are characterized by large changes in methylation” is a reasonable one that accurately reflects our observations.

Response Figure 3. Locus plots for DMRs identified at *MOV10L1* and *FZD6*, showing the results of 450k array analysis in the DMR carriers versus controls.

We have now edited the sentence to read as follows:

“Allelic analysis demonstrated that these epivariations generally represent large methylation changes specifically on one allele, with most showing two clusters of methylated and unmethylated reads occurring in approximately equal proportions (Fig. 1).”

(f) “We also observed that many hypermethylated epimutations at promoters are associated with complete silencing of one allele (Extended Data Fig. 7), and, thus, have an impact comparable to that of loss-of-function coding mutations” It is true that Extended data fig 7 shows an association between differently-methylated alleles and allele-specific expression in one individual. This does not demonstrate a causative relationship. Experimental demonstration is required for the authors to make the claim they do. In my opinion, as a result, the closing statement “Our study shows ... that epimutations ... often exert strong functional effects on gene expression” is not justified – in particularly the qualifier ‘often’.

We agree that the chosen examples shown in Figure 5A-E (previously Figure 4) do not demonstrate causality of an epivariation underlying unusual gene expression patterns. However, the relationship shown in Figure 5F, which includes data for 138 different epivariations, does show a significant relationship, where the presence of an epivariation at

gene promoters is associated with altered gene expression ($p=9.2 \times 10^{-5}$). While it can be argued this observation still does not demonstrate causality, the association of altered promoter methylation with altered gene expression is highly statistically significant, thus indicating a clear relationship between the two observations.

Of the gains of methylation listed in Supplementary Table 9, we would like to highlight that 17 occurred at the promoters of genes that also showed the lowest expression rank in the population, representing an 11-fold increase for extremely low expression outliers compared to that expected by chance. Where allele-specific measurements could be made, 26% of genes with autosomal promoter DMRs had allelic ratios $>4:1$, thus indicating highly biased allelic expression was a frequent occurrence. We have now edited the initial description in the main text to read as follows:

“We verified that epivariations encompassing gene promoters were often associated with changes in gene expression, with hypomethylation leading to increased expression and hypermethylation to transcriptional repression ($p=9.2 \times 10^{-5}$, Mann Whitney test) (Fig. 4). We also observed that some hypermethylated epivariations at promoters are associated with complete silencing of one allele (Extended Data Fig. 7), and, thus, have an impact comparable to that of loss-of-function coding mutations.”

The phrase “and often exert strong functional effects on gene expression” has also been removed from the concluding paragraph.

4. Lines 138-143 are not clear to me. What I am taking from these words is that apparently healthy individuals and individuals with ND/CA both harbour variations in DNA methylation, with a degree of enrichment on a purely numerical basis in the individuals with ND/CA, and an interesting question about whether this is related to their clinical condition. Can the authors clarify their claim here?

The reviewer is correct. Our testing of 2,711 samples from the general population (who were also compared against the same core set of 1,534 controls that were used to identify epivariations in our probands using an identical methodology), identified many DMRs in this population. These DMRs are listed in Supplementary Table 3. Similarly, Supplementary Table 2 also lists DMRs detected in 117 families taken from the general population. We found the frequency of DMRs identified in the 2,711 controls was slightly lower than that identified in patients with ID/CA. Thus, just as rare SNVs and CNVs are found in all of us, rare epivariations are also present in many individuals, and are not unique to disease patients. In accordance with comments made by Reviewer 3, we have now edited this section to specifically mention this, and that the pathogenic significance of many epivariations is unclear, such that is now reads as follows:

“Epivariations were identified in population controls, and some also occurred in apparently unaffected parents. 33 of the 57 DMRs identified in probands for which we investigated inheritance were also present in one parent, and we identified a total of 719 DMRs in the 3,326 control samples analyzed (Supplementary Tables 2 and 3). 24 of the epivariations identified in our cases were also present in one or more of these controls, suggesting either that these DMRs are unrelated to the patient phenotype, or perhaps are associated with incomplete penetrance. We observed a 1.2 fold enrichment in the frequency of epivariations in the 489 ND/CA samples when compared to 2,711 population controls (Extended Data Fig. 2), although this does not reach statistical significance ($p=0.058$, Fisher’s exact test). Testing of parental samples of 57 ND/CA probands showed that 42% ($n=24$) of the epivariations were *de novo* events. When compared to epivariations found in 117 control pedigrees⁷ (Supplementary Table

2), this represents a 2.8-fold enrichment in the rate of *de novo* epivariations in cases compared to controls ($p=0.007$, Fisher's exact test) (Fig. 2). Thus, while the pathogenic significance of many epivariations is unclear, the paradigm of *de novo* mutational events echoes that observed for other classes of genetic mutation (copy number and single nucleotide variation) deemed pathogenic in ND and CHD cohorts^{9,10}."

5. I am having some problems with the authors' view on the clinical significance of methylation changes in their case cohort.

As mentioned above, in 58% of samples where parents were available, epivariations were shown to be present in apparently healthy parents. Perhaps the discussion can explore the likely prevalence of methylation variants in the population, as has been done for CNVs.

Was any evidence sought or found that shared clinical features (e.g. the heart disease mentioned in a significant number of the case cohort) were associated with shared epigenetic changes?

"three epimutations encompassed the promoters of genes with prior known disease associations and/or hypermethylated triplet repeat expansions (MEG3, FMR1, FRA10AC1), validating our method for detecting pathogenic epimutations" (lines 155-158). This feels like a moment of linguistic sleight of hand, suggesting clinical validation of pathogenicity. As it happens, one of the individuals with a MEG3 change does have clinical features in common with an imprinting disorder. But since the authors don't in any way address specific clinical cases, it is important that they should not make claims that might be taken to indicate clinical validity.

As stated above, we have now added wording to clearly state that epivariations were also observed in population controls and in apparently unaffected parents, and as such it is unclear if these are relevant to the patient phenotype. We have also modified the concluding paragraph to state that these may be involved in human phenotypes, such that it now read as follows:

"Our study shows for the first time that epivariations are a relatively common feature in the human genome, that some are associated with changes in local gene expression, and raises the possibility that they may be implicated in the etiology of developmental disorders."

In the small number of cases where we observed recurrent epimutations found only in disease samples, we do already comment on the known examples of *FMR1* and *MEG3*, and then specifically mention the example of *MOV10L1*, which, based on its known function, represents an excellent candidate gene for congenital heart defects.

"This represents a significant enrichment for rare SNVs disrupting CTCF binding sites ... when compared to the same regions in controls" (lines 176-178). I am not clear from the text that the authors sequenced their control cohort in the same way they sequenced their cases. Can this be made explicit, so this significant enrichment can be justifiably stated?

The controls used in this test were the other unrelated parental samples sequenced with the same custom capture panel but who did not carry epivariations at these loci (here we did not include offspring, as to do so would mean we were simply re-sampling the same haplotypes as present in the parental samples). Thus, the sequencing method was identical to that used to assay the individuals who carried CTCF-disrupting variants, making this a reasonable comparison. However, the reviewer is correct in that we did not make this clear in the text. We have now edited this sentence such that it reads as follows:

“This represents a significant enrichment for rare SNVs disrupting CTCF binding sites in the vicinity of epivariations when compared to the same regions in other sequenced samples who did not carry epivariations at these loci ($p=0.0015$, Fisher’s exact test), strongly implicating rare *cis*-linked variants in regulatory sequence as a causative factor underlying some epivariations.”

“Our study shows for the first time that epimutations are ... implicated in developmental disorders” (lines 247-248). This claim is not justified.

We have edited this sentence such that it now reads:

“Our study shows for the first time that epivariations are a relatively common feature in the human genome, that some are associated with changes in local gene expression and raises the possibility that they may be implicated in the etiology of developmental disorders.”

6. I am not finding the data related to the assertion in lines 234-245. It feels rather unprepared, unsubstantiated and isolated tucked in just before the discussion section, and doesn't seem to fit in with the rest of the data; perhaps it could be omitted.

The reviewer is correct in that we omitted to reference the relevant piece of supporting data at this point in the text (Supplementary Table 2). A reference to this table, which contains a full breakdown of the heritability analysis discussed, has now been added. We have also added additional text to this section to more fully explain the hypothesis being tested.

We think that this is an important observation to retain in the manuscript, as it provides some novel insight into the basic biology of epivariations, and acts to affirm the dynamic nature of the events we detect. In addition, it also acts as an important counterpoint to the high rate of *de novo* occurrence identified earlier in the study, thus giving some explanation as to why epivariations exist at overall relatively low levels.

Reviewer #2 (Remarks to the Author):

The paper by Barbosa et al is very exciting. Although set in the context of NAs and NDs the paper is relevant to a vast amount of research that is being conducted to understand the genetic aetiology of human disease/disorders that is almost universally falling short of its ambition. This paper explores an alternative with considerable and optimistic outcomes. The resources and context of the study are excellent.

Some care needs to be given to language in this manuscript.

Epimutation

The term “epimutation” is strictly defined as a heritable change in gene activity that is not associated with a DNA mutation but rather with gain or loss of DNA methylation or other heritable modification of chromatin. Changes in gene expression through altered DNA methylation or histone modifications induced from cis- or trans-acting genetic factors known as methylation Quantitative Trait Loci, (mQTL) are therefore not epimutations in this strict sense. Please modify the text accordingly.

As per the suggestion of Reviewer 1, we have modified our usage of the term epimutation to “epivariation” throughout the manuscript, including the title.

In some instances the manuscript is difficult to follow without extensive reference to the Supplementary material. Could the authors provide further information around the

**following aspects to improve the flow of the presentation without extending the text unreasonably? Eg
After stringent quality control... line 109**

We have added a brief summary of how we performed QC. The sentence now reads as follows:

“After stringent quality control, including removal of loci with clusters of poorly hybridizing probes and extensive manual curation, we identified a total of 143 epivariations in 114 ND/CA samples (*i.e.*, 23% of the probands tested).”

Each sample was screened for epimutations...line 107

We have added a brief summary of how we identified epivariations. The sentence now reads as follows:

“We utilized a sliding window approach to identify epivariations in each sample, defined as 1-kb regions containing ≥ 3 probes showing strong outlier methylation absent in the 1,534 controls (see Extended Data Fig. 1 and Online Methods section).”

Secondary validation an assessment....line 112

We have re-written the sentence to state this up front so that it is clearer that bisulfite sequencing was the method used, also incorporating comments from Review 1. The sentence now reads as follows:

“Using PCR/bisulfite sequencing, we performed orthogonal confirmation for 70 epivariations (Supplementary Table 5), yielding a 95% true positive rate.”

The brief methods state that an “epimutation” (see above) is defined as strong outlier differentially methylated regions absent in the 1,534 controls – how were the outlying control (as described and shown in Figure 1) managed in the analysis?

We deliberately set our DMR calling criteria in such a way as to allow some flexibility to allow for the presence of outlier probes in single controls. If we had not done this, it would otherwise mean that the presence of just one outlier probe found in a single control sample could mean that we could never identify a DMR in that region. As outliers can be caused by hybridization artifacts or the presence of rare variants at the CpG or probe binding site, this was an important consideration in our analysis pipeline.

To build in this flexibility to account for outlier probes in controls, we required that a DMR contained at least 3 probes that were all within the 0.1% tail of the distribution of control beta values. Given that we have >1,000 controls, this means that at any probe position, it is possible for a DMR to be called in a case at that locus even if there is one control sample that has a more extreme methylation value than the case sample (although note that a DMR has to contain one or more probes that have beta values >0.1 more extreme than that seen in all controls).

We also performed extensive QC on the controls utilized and removed samples that were outliers in various ways. First, using PCA plots, we removed control samples that were outliers from the main group of each GEO entry. Secondly, we also performed DMR calling on each control sample versus the rest of the control group, removing any controls that reported >10 DMRs.

Figure 2. I think this would be better presented as total n rather than percentage. Please

also check the numbers quoted – are de novo “epimutations” n = 49 (legend) or n = 24 (Figure)? Are the number of de novo “epimutations” in controls 6 (figure) or 40 (legend)?

The numbers shown within Figure 2 are correct and show actual numbers, not percentages. However, the reviewer correctly points out there was an error in the legend to Figure 2, in that we state 49 epivariations were found in cases, when the correct number is in fact 57 (24+33). We apologize for this error, have corrected it in the legend, and also modified the legend text to improve clarity, as follows:

“Figure 2 | A significant excess of *de novo* epivariations found in patients with ND/CA. We observed a 2.8 fold enrichment for *de novo* epivariations in cases (n=24 out of 57) when compared to controls (n=6 out of 40) (p=0.007, Fisher’s exact test).”

Figure 3. There are two green lines in this figure – could the colour coding be changed?

We suspect that the reviewer was referring to the two green bars shown in the UCSC screenshot shown in the lower panel of Figure 3A (there was only one green line in Figure 3). To avoid confusion, we have changed the color of the green bar that shows the position of the CTCF motif within with ChIPseq peak to blue and also modified the legend to make it more explicit, as follows:

“In the lower the UCSC Genome Browser view, the DMR location is shown in as a green bar, and the a rare SNV that lies within the CTCF binding motif (blue region within black bar) in this same individual is shown in red.”

What does “often strongly” mean at line 203?

The full sentence here is “We verified that epivariations encompassing gene promoters often strongly impact gene expression, with hypomethylation leading to increased expression and hypermethylation to transcriptional repression (p=9.2x10⁻⁵, Mann Whitney test) (Fig. 4).” In this context, we meant it to state that when DMRs were located at gene promoters, we often observed a large change in the expression level of the associated gene. Two examples are shown in Figure 4, where the DMR carrier shows the lowest or highest expression out of 462 individuals assayed. However, we recognize that this wording is perhaps unclear, and have modified the statement to read as follows:

“We verified that epivariations encompassing gene promoters were often associated with large changes in gene expression”

I do not think that CTCF is defined anywhere in the main text.

We have added CTCF’s full name (CCCTC-binding factor) at the first mention in the text.

Line 68 and 94: What types of microarrays are the authors referring to and what are they measuring?

We were referring to the various types of microarrays used to detect CNVs. To clarify this statement and distinguish from other types of arrays (e.g., expression arrays), we have modified the text to read “CNV microarray”.

Reviewer #3 (Remarks to the Author):

Barbosa et al. perform a deep and comprehensive analysis to identify DMR patterns in ND/CA pro-bands and characterize them in a most detailed fashion. We find this study covers the subject very extensively, and reveals possible correlation between epigenetic patterns, gene expression and the ND/CA. We have raised several issues that should be considered and/or explained before this manuscript is considered for publication. Below are more major comments, followed by minor ones:

1. CTCF participates in methylation activity, thus the canonical splice site SNVs and CTCF's regulation function can definitely explain the DMRs in that vicinity. A disrupted CTCF protein could also correlate with methylation dysregulation throughout the genome and by that explain the enrichment of DMRs of those specific cases. Can this bias the overall DMR enrichment in cases (observed in line 142?)

We do not report any splice site SNVs, so I am not quite sure why this is mentioned – perhaps the reviewer meant to instead write TFBS SNV? We agree that disrupting a single binding site for CTCF could feasibly result in local epigenetic dysregulation, and indeed this hypothesis that the rare SNVs we detect at CTCF binding sites are likely the underlying cause of why we observe a DMR at that locus is the one that we favor. However, given that we detect only three samples with SNVs that disrupt CTCF sites in a total of 489 samples tested, these 3 DMRs do not account for overall the enrichment of DMRs we observe in cases. Furthermore, as each individual with disruption of a CTCF binding site had only a single DMR identified in their genome, we observed no evidence for a wider disturbance of methylation in their genomes.

2. Are the 70 samples the only ones that were validated using bi-sulfite? (line 113)? Why were those samples selected and how was the selection process performed? Also, if we understand correctly, 24/70 epimutations were verified as de-novo events (34%, extended data line 143-146). The authors point this as 42%(??). Furthermore, this seems like a very high proportion of new to inherited events. How do the authors explain this number? What is the proportion to be expected from literature? On the same subject, 117 control families were used to assess denovo rate in controls. What method was used to call the denovo epimutations in the controls?

The fraction of *de novo* DMRs, as shown in Figure 2, is 24 out of 57 for which inheritance could be determined, yielding the number quoted of 42%. Although line 112/113 states “Secondary validation and assessment of inheritance for 70 epivariations was performed...”, for some of these 70 loci only probands were tested and, thus, although the DMR was validated, no inheritance information was available. Further, some of the 70 loci failed to validate (*i.e.*, the DMR call from the array was likely a false positive), bringing down the number of true positives for which both parents were also available down to 57. These results are shown in Supplementary Table 5. To clarify, we have now edited this sentence such that it now refers only orthogonal validations. It now reads:

“Using PCR/bisulfite sequencing, we performed orthogonal confirmation for 70 epivariations (Supplementary Table 5), yielding a 95% true positive rate.”

A later sentence on inheritance of DMRs reads:

“Testing of parental samples of 57 ND/CA probands showed that 42% (n=24) of the epivariations were *de novo* events.”

There is very little prior literature in this field, and thus there is little precedent for the fraction of *de novo* events that we could expect to observe. We agree though that the proportion of *de novo* DMRs we observed in cases is very high, and we found this a surprising finding. This is why we then went on to characterize the rate of *de novo* versus inherited DMRs in the normal population, using 117 families published by McRae *et al.* We used an identical pipeline for calling DMRs in this control cohort as was used in the cases. After DMRs were called, we determined their inheritance within each family by making a single plot of the DMR showing beta values for all individuals in the pedigree, and visually ascertaining which family members carried the DMR vs which had a normal methylation pattern. This was a much more robust approach than simply using the automated DMR calls coming from our script, as in some instances small fluctuations in beta values among individuals meant that even though a child and one parent both clearly had outlier beta values and thus the DMR was an inherited event, the methylation values might only reach the thresholds to be formally called a DMR in one family member.

3. How did the authors test for significance of enrichment in cases vs. controls? How did they correct for multiple testing deriving from multiple regions in the microarray? This is also relevant for the CTCF enrichment in line 177 since 50 other DMRs were also tested. Regarding the SNVs in CTCF: what was the number of control variants in this location? What is the population allele frequency in general population? (ExAC/gnomAD?)

As stated on line 143 of the manuscript, significance testing for the proportion of *de novo* DMRs in cases versus controls was performed using a Fisher’s exact test. In this case, the test does not require a multiple testing correction as we are simply testing a single frequency (the number of *de novo* DMRs in cases versus the number of *de novo* DMRs controls).

We agree that if we had tested whether multiple transcription factors binding sites were disrupted in cases versus controls, then a multiple testing correction would be appropriate. However, this was not the case. In fact, CTCF was the only transcription factor binding site that we observed to be as disrupted by rare SNVs in >1 sample (Supplementary Table 12), and, therefore, we only performed an enrichment test for this single factor. As such, no multiple testing correction is necessary here.

Supplementary Table 12 already lists the allele frequency in the 1000 genomes population of each SNV. However, we have now added an additional column that shows the allele frequency in the gnomAD database. A few variants listed in the original version of the table had gnomAD allele frequencies >1%, and these have now been removed.

Regarding SNVs and CTCF binding sites, there were 3 DMRs found in probands that were associated with SNVs disrupting a CTCF binding site. Across the three sequenced regions surrounding the DMRs, each of which spanned ~75 kb, we identified a total of 1,888 SNVs and small indels in the 139 samples sequenced. Seven of these variants overlapped annotated CTCF binding sites based on ENCODE Factorbook annotations, representing CTCF consensus binding motifs located within CTCF ChIPseq peaks. Three of the seven variants were identified in probands with a DMR at that same locus (*i.e.*, the two events co-occurred together), while four CTCF variants were identified in other samples who did not carry a DMR of the locus. The three CTCF variants found in probands with a DMR all occurred in very close proximity to the DMRs (<1 kb separating the DMR and the mutated CTCF site). In contrast, the CTCF variants found in other samples (individuals without a DMR at that locus) all occurred distant to the location of the DMRs (>10 kb). This co-localization of DMR and TFBS-variant provides good (albeit circumstantial) evidence of a causal link between the two events. We have now added

text to state that all three cases of CTCF binding site mutations occurred in close proximity to a DMR.

4. How did the authors correct for gender? cases have 68% males and controls 40%.

As stated in the reply to Reviewer 1, we did not perform any correction for gender. Although the gender proportions in cases and controls are not the same, our control cohort contains hundreds of individuals of each gender. Therefore, sites that show differential methylation according to gender will be very well represented in our control cohort. As our method for identifying DMRs then looks for regions where methylation levels are unique to a single case and never observed in controls, the fact that the gender ratios are not equal in case and control population is not a significant factor.

5. The average age for cases – 10 years, controls is 56 years old. Methylation signatures are being accumulated with age, thus the test of young to old makes less sense.

Again, we did not apply any correction for age, as even though the mean ages of case and control are different, it is not necessary to apply any correction given the type of test we are doing. Our analysis for identifying DMRs compares the methylation profile of a single individual to the complete spectrum of methylation levels observed in all controls. DMRs are only called where the beta value in a case is either >0.1 above the maximum observed in all controls, or >0.1 less than the minimum observed in all controls. As our control population includes dozens of children, even though the mean age of all controls may be different to the mean age of all cases, the control group ages cover the full age range of cases. Thus, as for gender, the fact that the ages of case and control are not matched is not a significant factor in our DMR calling method.

6. GSE55763 is comprised of 2,664 controls and 36 technical replicates, but authors claim they used 2,711 controls from that dataset. Also, what is the nature of these controls? Do they belong to a specific age/phenotypic group that might affect their methylation patterns?

We downloaded the data for GSE55763 from GEO (<https://www.ncbi.nlm.nih.gov/geo/query/acc.cgi?acc=GSE55763>). Even though the associated abstract states that there are 2,664 controls and 36 technical replicates, this accession actually contains data from 2,711 arrays. While some are technical replicates, we utilized all hybridizations in our analysis (the replicates represent only ~1% of the entire data set). According to the associated publication (PubMed ID: 25853392), this cohort is comprised of 1,080 Type 2 Diabetes cases and 1,607 controls. We are not aware that Type 2 diabetes is known to have a sufficiently distinct methylation profile that would lead to significantly different results given the nature of our DMR calling pipeline. Also, as our control population comprises 1,534 samples, given the high frequency of Type 2 diabetes in the general population, it is reasonable to assume that some of our 1,534 controls are also diabetic.

7. What are the criteria to decide on an epimutation (line 110)? DMRs were rated visually by two researchers (Extended data line 100). What were the parameters by which each decided on a true positive DMR? What was considered as false positive?

This is an important question. All DMRs were manually curated to remove loci that were deemed false-positive calls. We observed several types of false positive that were removed:

(i) In some cases while 3 probes within a 1-kb region were called as outliers, these outlier probes were not contiguous as we would expect for a true methylation change, and were interspersed with other probes that showed no difference compared to the control population. We interpreted these signals as likely random groupings of individual probes that each yielded outlier beta values for some other reason, *e.g.*, rare underlying variations that influenced probe performance, or poor hybridization performance of individual probes. Indeed, we identified that some such cases were due to regions of homozygous deletion as indicated by clusters of probes with failed array detection p-values. In this scenario, probes have no target to bind to, and will, therefore, report essentially random beta values, albeit with very low signal intensity. It was this realization that prompted us to incorporate a filtering step to try and remove these regions represented by clusters of probes with poor detection p-values (Supplementary Table 10). The figure below shows an example of a locus that was removed due to this reason, with multiple outlier probes and multiple missing data points that were removed due to poor detection p-values (low signal intensity). While this region was called algorithmically as a DMR, it was removed during manual curation.

Response Figure 4. Example DMR called algorithmically that were deemed false positives during manual curation due to multiple missing data points. This region shows multiple outlier probes and multiple missing data points that were removed due to poor detection p-values (low signal intensity). While this region was called algorithmically as a DMR, it was removed during manual curation.

(ii) In other cases, batch effects, *i.e.*, technical differences due to arrays being processed in separate groups, were presumed to be the cause of signal differences between cases and controls. This phenomenon was the most prevalent. Here, it was usually observed that there was either a systematic shift in the beta values reported by one or more probes within a region between arrays processed in different batches. In some cases the mean of each batch was significantly different, with every sample showing a shift, whereas in other cases the means of the two populations remained similar, but a subset of the samples in one batch showed a gradient of deviations, with the beta values of multiple cases lying in the extreme tail of the control distribution. Examples of both are shown below. We have added a statement outlining these two types of false positive events in the Supplementary Methods.

Response Figure 5. Example DMRs called algorithmically that were deemed false positives during manual curation. (left) The individual with the putative DMR is shown in red. The distribution of 1,534 controls is shown as shades of grey corresponding to ± 1 , ± 1.5 and ± 2 standard deviations from the control population mean, which is represented by the dashed black line. Dashed grey lines represent controls with outlier methylation levels. In this instance, it can be seen that there are many control individuals that show a gradient of methylation levels that are >2 Standard Devs above the mean. While the case called with the DMR simply has a slightly more extreme methylation value than all the controls, it is not very different to that seen in some of the more extreme controls. Because of this, in this example we deemed that the case methylation profile was not a sufficiently convincing outlier to be called a true positive. **(right)** Here the individual with a DMR called algorithmically is shown in green. The dotted red line shows that mean of the cases tested in this batch of arrays, while the dotted black line with expanding shades of grey shows the control mean and 1, 1.5 and 2StDevs of the control mean. Here it can be clearly seen that at this locus there is a significant batch effect, as the mean of cases and controls is very different. As such, we removed this DMR as a false positive due to a systematic batch effect.

8. While nicely depicted, the effect of hypo/hyper methylation of promotor regions (lines 200-208) has long been known and discussed. What is the benefit of this analysis to what was previously known? The authors should recognize previous studies describing this phenomenon. Also, the correlation between methylation patterns between tissues (lines 223-232) has also been deeply described previously. For example, see: Lokk et al, 2014, Genome Biology, or the more recent Guo et al, Nat Genetics, 2017. What is the benefit of this study compared to what was previously described?

We are aware that there is an extensive literature that already demonstrates a negative correlation between promoter DNA methylation and transcription, and we do not claim that our demonstration of an association between change in DNA methylation and changes in gene expression is novel. However, as our manuscript represents the first deep description of epivariations in the human population, we have attempted to provide as comprehensive a description of their occurrence and effects as possible. In this context, we think that our characterization of altered gene expression patterns associated with the presence of epivariations represents important observations to include in this manuscript, as they show that the DMRs we detect are associated with functional effects on the genome. Without performing this analysis, there would remain a major question as to whether the methylation changes we detect are functionally important or merely inconsequential epigenetic trivia observed in some peoples genomes. Further, we have now referenced a review article that discusses the known links between DNA methylation and gene expression.

Similarly, we are also aware that previous studies have described that, in some instances, methylation patterns do show strong similarity across tissues. However, it is also well known that different tissues can often display quite distinct methylation patterns. For example, the paper of Lokk *et al.* quoted by the reviewer, entitled “DNA methylome profiling of human tissues identifies global and tissue-specific methylation patterns” describes sites of both similarity and difference among tissues, including many tissue-specific DMRs. We considered it particularly important in our analysis to investigate whether the rare DMRs we identified tended to be found in multiple tissues, as it was possible that these represented either somatic or germline events. Also, as we tested DNA derived from blood, their potential pathogenic

significance would be very different if such events were not present in multiple other tissues in carrier individuals. We have added a reference to the Lokk *et al.* paper to the manuscript.

9. Also regarding gene expression changes with regards to methylation patterns: we could not find the relevant information regarding the numbers of genes that are hypo/hyper methylated and their expression differences.

Supplementary Table 9 contains a complete list of all DMRs and associated genes that were analyzed. This table forms the basis for Figure 4F that summarizes the effect of promoter DMRs on gene expression.

10. Figure 1 - There's no legend and no text to relate to the various panels.

We are rather confused by this comment, as the manuscript we uploaded contains a detailed legend to Figure 1, including a description of panels A, B, C and D.

11. Figure 1 panel A – if parents are 50% methylated and child is 20%, why is that considered De-novo?

Figure 1 shows the imprinted locus *MEG3*. In normal individuals, this region is highly methylated on the paternally-derived allele, and unmethylated on the maternal allele. As such, normal individuals show ~50% methylation. The two cases shown in green (Proband 146 and 398) both show a clear loss of methylation that is significantly less than the lowest methylation values seen in any of the controls. Testing of the parents of Proband 398 (blue and red lines) shows that both carry the normal methylation pattern (close to 50%) at this locus, and therefore the loss of imprinting observed in Proband 398 must be a *de novo* event where the paternal allele they carry has largely lost methylation. We agree that the methylation levels in the two children are not zero, but this is a common finding seen in most cases where imprinted loci lose methylation (for examples, please see our previous publication on methylation anomalies in patients with uniparental disomy, PubMed ID 27569549). In part, my belief is that it may be due to technical performance of the Illumina 450k array, as if one uses bisulfite sequencing of these loci one will generally observe that all reads are unmethylated, even though they array results might suggest 10-20% methylation remains.

12. No explanation of the regions that were chosen for panels A-C in the text

We chose to show *MEG3* and *MOV10L1* as both were recurrent events observed in two unrelated probands. As such, they have a higher weight of evidence indicating that they are likely to be related to the patient phenotypes. *ZNF57* was chosen for panel C/D as this was one of the few genes where we identified an informative SNP in the bisulfite amplicon that allowed us to unambiguously show the gain of methylation occurred specifically on one allele (the data presented in the lower part of Figure 1D).

13. *MOV10L1* (hg19: chr22:50528178-50528751): this region is not just the promoter. it covers the 5' UTR as well as first exon and part of intron

The reviewer makes a good point. We have edited the text to state this, and it now reads as follows:

“Recurrent hypomethylation at the promoter/5' UTR/first exon of *MOV10L1*”.

14. We are uncomfortable with the use of the CADD score for ##. Several recent studies show CADD to have large amounts of false positive variants (low specificity) in non-coding regions (Gelfman et al. 2017, Mather et al. 2016, van der Velde et al. 2017, Shihab et al. 2015, etc). Some of the other scores present significantly better tp/fp rates in direct comparison with CADD for various non-coding annotations.

The original version of Supplementary Table 12 does include a column of CADD scores for each SNV identified. However, these were included only as additional annotation, and at no point do we use these scores in the manuscript. We have therefore removed this column from the revised table included in the current submission.

15. What is the difference in calling methods for epimutations between cases and the different sets used for controls?

We used an identical method for calling DMRs in cases and controls, *i.e.*, there is no difference. Both were compared against the same core set of 1,534 control samples, all samples were normalized together as a single batch, and both cases and controls used the exact same DMR calling thresholds.

16. What is the rate of recurring epimutations in controls? How many controls share the same epimutations as were shared between cases (lines 156-157)

The reviewer raises a good point that we did not adequately address in the initial submission. In fact, cross checking the list of DMRs identified in cases and our two population of controls (117 families, and 2,711 unrelated population controls), we observed that 24 of the epimutations identified in our cases were also present in one or more of these controls. This included five of the 12 recurrent epimutations, and also two losses of methylation found at the imprinted loci *NAA60/ZNF597* and *L3MBTL1*. Clearly this is of importance when considering their potential pathogenic significance, and thus we have now added statements at several points in the text to reflect this. The relevant sections now read as follows:

“Epimutations were identified in population controls, and some also occurred in apparently unaffected parents. Thirty-three of the 57 DMRs identified in probands for which we investigated inheritance were also present in one parent, and we identified a total of 719 DMRs in the 3,326 control samples analyzed (Supplementary Tables 2 and 3). Twenty-four of the epimutations identified in our cases were also present in one or more of these controls, suggesting either that these DMRs are unrelated to the patient phenotype or, perhaps, are associated with variable penetrance. We observed a 1.2 fold enrichment in the frequency of epimutations in the 489 ND/CA samples when compared to 2,711 population controls (Extended Data Fig. 2), although this does not reach statistical significance ($p=0.058$, Fisher’s exact test).”

“We identified 12 recurrent epimutations (Extended Data Fig. 3), *i.e.*, the same methylation change was identified in multiple unrelated probands. Of these, two epimutations encompassed the promoters of genes with known disease associations (*MEG3* and *FMR1*)^{6,11}, validating our method for detecting pathogenic epimutations. A third recurrent epimutation coincides with a locus containing a hypermethylated triplet repeat expansion (*FRA10AC1*)¹² although this, and four other recurrent epimutations detected in our disease cohort, were also identified in population controls, suggesting that they are unlikely to be pathogenic. One of the novel recurrent epimutations detected only in our patient cohort was found in two patients with CHD (Probands 22 and 117). This recurrent hypo-methylation defect at the promoter/5’ UTR/first

exon of *MOV10L1*, a gene with an embryonic heart-specific isoform that interacts with the master cardiac transcription factor NKX2.5¹³ (Fig. 1).”

“Of note, we observed loss of methylation at two known imprinted loci that have no prior disease associations (*NAA60/ZNF597* in Proband 6 and 62, and *L3MBTL1* in Proband 308), although in both cases similar losses of methylation were also observed in population controls, making the pathogenic significance of loss of imprinting at these loci unclear.”

17. How were the specific 50 DMRs chosen for targeted sequencing? (line 170)

The DMRs chosen for targeted sequencing were done based on purely practical reasons. First, we were limited to those samples for which we had sufficient DNA remaining after array testing and bisulfite PCR validation, and preferentially those for which DNA from both parents were available, as interpretation of variation without parental samples is difficult. Secondly, the targeted sequencing required us to design and order a custom sequence capture library, meaning that the content was fixed once we placed the order. At the time that we made this custom capture design, we had only completed analysis of some of samples that are described in this manuscript, and at that point we simply included every DMR that we had identified at the time which had validated, and which we thought we could successfully sequence given the DNA samples available to us.

18. Extended Data Figure 4: where are the Bisulfite sequencing validations of the proband for HM13. Also, legend read both empty and full circles as “Methylated CpG”

Unfortunately after completing array analysis we did not have any remaining DNA to perform bisulfite validations for the proband with a gain of methylation at *HM13*, and, thus, we were unable to generate bisulfite sequencing data in this sample. We have added a statement of this to the figure legend. However, we still thought it useful to show data from the parents. Thank you for pointing out the error in the figure legend, we have now corrected this.

19. Also, there is no chance to read the text in Extended figure 9 to know which graph is which tissue.

This may be related to the fact that Supplementary Figures are often shared with reviewers at lower resolution than they appear in the published version. However, to ensure that the labels are legible, we have enlarged the font of the panel labels to more clearly indicate which tissue they are derived from.

20. Extended Data Figure 5: looks very messy, should this be divided into panel A+B in one figure, C and D as separate figures?

In order to make the figure cleaner and to more clearly show which panels relate to each other, we have now added borders around each of panels A, B, C and D in order to visually group together the relevant sub-parts of the figure. We believe doing so makes the figure clearer for the reader, without the need to create additional figures. Also, we believe that these data make sense to be grouped together as they all address the same point, *i.e.*, small CNVs identified at or close to epivariations.

21. Extended Data Figure 6: The result is written in the legend of the figure

Given the space limitations of the manuscript main text, we deliberately wrote the legend to be fully explanatory. Doing so allows this figure to be easily understood by the reader, without having to cross-reference with other parts of the manuscript. If the editors would prefer that this information is moved to the main text, we would be happy to do so.

Reviewer #1 (Remarks to the Author):

Barbosa et al revision

This reviewer thanks the authors for a comprehensive rebuttal letter, and accompanying changes to the text that address many comments. Since the rebuttal and changes are quite extensive (and not consistently numbered and tracked), I will attempt to follow the format of the initial rebuttal. I will only mention rebuttal points where I have residual questions.

Reviewer 1

1.1: the authors have answered, but not addressed, the comment about correlating their variations with the clinical presentations of the probands.

1.2a: the statistical criteria are now explained. It is not explicit whether the "stringent" thresholds involved consecutive probes with altered methylation. From other comments in the rebuttal this appears to be so but it would be nice to have this clarified.

1.2b: this answer deals with the identification of "true positives" based on "manual curation".

(i) Normally I regard a 'true positive' result as one orthogonally demonstrated by a gold standard method - in this case, this should be verification of a methylation change by bisulphite sequencing. However, the authors use true positive to define a methylation array results after manual curation.

(ii) Manual curation, as well explained here, was chiefly used to weed out results assumed to arise from low signal intensity, and from batch bias. I am quite surprised that the first of these was not achieved during initial QC. The second of these, batch correction, was the subject of my Q2d. I remain a little surprised that these corrections were not applied informatically in initial processing. I am also unsure how any potential bias in interpretation can be applied to manual curation of case and control samples. Were the manual curators blinded to the samples?

1.2c: it's great to have this method clarified. I am still quite surprised that "extreme methylation values" were predicated as outside the furthest control outlier, rather than statistically.

1.2d: the rebuttal to my question here re-states some of my remaining uncertainties listed above.

1.3b: I still don't like DMR as a term. I guess we agree to differ here.

1.3e, and response figure 2. I was surprised that some of the DMRs presented here are shown to have average methylation difference between 0 and 0.1. I understood from response 2a that the stringent criteria require an average methylation difference ≥ 0.1 .

I was a bit disconcerted to read that the bisulphite sequencing and 450k results (as presented in Fig 5) are discrepant, and the authors surmise that the 450k results are more valid. As mentioned above (my remark 1.2.b(i)) I believed the gold standard here to be bisulphite sequencing, rather than 450k. It appears that some of the 'large changes' depend on the method used to measure them.

1.3.f: The revised text of the authors refers to methylation: transcription correlations from a dataset of 'normal' people. I still feel that to state these 'have an impact comparable to that of loss-of-function coding mutations' is to suggest an inference that is not present in the data. The authors are extrapolating from methylation analysis of patients to expression analysis of different genes in different, healthy, people. It should be very clear that this is what they are doing.

1.4: This revised text is awkward and unclear; it has the feel of a quick fix. It also results in pretty much the first statement of the paper being that methylation changes occur in probands, parents and controls and aren't significant, so that when the original text continues talking about significance, it is a non sequitur.

1.5: Two of the three cited examples here (MEG3, FMR1) ARE associated with clinical disorders. They represent positive controls, as it were; and other publications have shown different ways of detecting them. They aren't relevant as stated here.

1.6: again the modified text is awkward and adds in clarity "the same regions in other sequenced samples" - how many? which samples? The authors have rebutted, but not quite fixed.

Reviewer 2 brings up manual curation and 'epimutation' calling again. In their rebuttal the authors state: 'We deliberately set our DMR calling criteria in such a way as to allow some flexibility to allow for the presence of outlier probes in single controls. If we had not done this, it would otherwise mean that the presence of just one outlier probe found in a single control sample could mean that we could never identify a DMR in that region'. This statement invites the suggestion that statistical criteria might have been more robust.

Reviewer 3

3.2: responding to questions about validation, the authors point out that some (10 of 70) methylation changes were not validated by bisulphite sequencing. This invites the comment that the text "we performed orthogonal confirmation for 70 epivariations (Supplementary Table 5), yielding a 95% true positive rate" is disingenuous, since the experiment was attempted on 70 but results secured on only 60.

3.3: I don't feel the authors directly addressed the comment 'How did the authors test for significance of enrichment in cases vs. controls' in CTCF sites. The response: "This co-localization of DMR and TFBS-variant provides good (albeit circumstantial) evidence of a causal link between the two events" implies no statistical validation.

3.11 and 3.12 are about MEG3, whose hypomethylation is a recognised association with Temple Syndrome. Given the established diagnostic and research activity for this methylation change, it's a shame the authors didn't secure validation for this assay, though they did present it in fig1.

3.16: I found this answer startling. How did the authors not previously determine the overlap of methylation changes between cases and controls? It brings me back to my original first comment: that 'the authors have not invested heavily in the clinical characteristics of their cohort or exploring the interplay between molecular and clinical interpretation'. The alterations to the text are a partial fix but I feel there could be more measured moderation of the tone and assertions of the paper.

Reviewer #2 (Remarks to the Author):

The authors have provided a constructive and thoughtful response to this review.

Edits to the manuscript have been made where necessary and I have no further comment.

Reviewer #3 (Remarks to the Author):

The authors have adequately addressed most of my comments. I believe this work presents a strong addition to the research of epigenetic variation as cause for disease and I therefore recommend this paper for publication.

Response to Reviewers Comments on revised manuscript, “Identification of rare *de novo* epigenetic variations in congenital disorders” by Barbosa et al.

Editors Remarks

In this case I would like to stress that even though we are interested in publishing your study, we consider this round of revision the last chance to alleviate all of Reviewer #1's remaining concerns. This will have to include a more in-depth consideration of the clinical aspects of the identified epivariations, a more sincere examination of the discrepancies in results between the methylation array and bisulfite sequencing and, thus, also moderation in the presentation of results.

In response to this guidance, we have made the following edits beyond those specifically requested by the reviewers to moderate the tone of the manuscript, as follows. In each case, all edits are made to the revised manuscript using the “Track Changes” function of Word, and any edited segments of text are shown in yellow highlights below.

More in-depth consideration of the clinical aspects of the identified epivariations

We have added additional text to summarize the frequency of clinical phenotypes in the overall test population, with reference to Supplementary Table 1, as follows:

Lines 96-100: “Almost 90% of the patients had a ND, 50% were classified as having an autism spectrum disorder, 16% had an epilepsy/seizure phenotype. 65% also had multiple CAs, including congenital heart defects (CHD) (36%), facial dysmorphisms (29%), growth anomalies (22%), and micro/macrocephaly (13%) (full details in Supplementary Table 1).”

We have added text to summarize the phenotypes of recurrent epivariation we detected specifically in cases, as follows:

Line 190-194: “The two males identified with hypermethylation at *FMR1* both had phenotypes consistent with a diagnosis of fragile X, primarily ID and behavioral anomalies. While both had previously been tested by PCR and reported as normal, subsequent Southern blot testing confirmed the presence of the classical *FMR1* triplet repeat expansion, although in one case this was an apparent mosaic event (data not shown).”

Line 201-203: “One patient with this epivariation at *MOV10L1* presented with double-outlet right ventricle, hypoplastic left ventricle, asplenia and short stature, while the second presented with pulmonary stenosis, laryngo-bronchio-tracheomalacia and foot polydactyly.”

More sincere examination of the discrepancies in results between the methylation array and bisulfite sequencing

1. We have edited the section that describes bisulfite PCR validation assays to make it explicit that 58 of the total of 70 attempted assays were successful. This section now reads as follows:

Lines 118-121: “Using PCR/bisulfite sequencing, we attempted orthogonal confirmation for 70 epivariations. We observed concordant changes in methylation for 55 of the 58 assays that

provided useful data, yielding a 95% true positive rate for DMRs detected by array (Supplementary Table 5).”

In addition, we have now added a new Supplementary Figure 3 that clearly shows our rationale for interpreting the results of bisulfite PCR/sequencing assays, and have revised the section of the manuscript where we discuss bisulfite validations to state that in some cases the interpretation of validation experiments was made complex due to highly biased allelic amplification. The section now reads as follows:

Lines 118-127: “Using PCR/bisulfite sequencing, we attempted orthogonal confirmation for 70 epivariations. We observed concordant changes in methylation for 55 of the 58 assays that provided useful data, yielding a 95% true positive rate for DMRs detected by array (Supplementary Table 5). Allelic analysis demonstrated that these epivariations represent large methylation changes specifically on one allele, consistent with the hypothesis that epivariations represent allelic events. In most cases we observed two clusters of largely methylated and unmethylated reads occurring in approximately equal proportions (Fig. 1), although in some instances the interpretation of validation experiments was made complex due to highly biased allelic representation, presumably reflecting preferential PCR amplification of one allele (Supplementary Figure 3).”

Moderation in the presentation of results

1. The last sentence of the abstract has been edited to remove any reference to the specific proportion of patients in whom epivariations might be causative and the relative diagnostic yield and to moderate the tone, and now reads as follows:

“We propose that epivariations contribute to the pathogenesis of some patients with unexplained ND/CAs, and as such likely have significant diagnostic relevance.”

2. Lines 157-160 we make a specific statement that epivariations are also found in controls, and therefore are not always pathogenic:

“In addition to searching for epivariations in samples with ID/CA, we also screened two large cohorts of population controls, identifying a total of 719 DMRs in the 3,326 control samples analyzed (Supplementary Tables 2 and 3). Thus, epivariations are a relatively common occurrence in the human genome, and are not always associated with any discernable clinical phenotype.”

3. Lines 156-160 have been moderated to specifically state that we are uncertain whether many of the epivariations are linked to patient phenotype. It now reads as follows:

“Thus, while it is currently unclear whether many of the epivariations identified contribute to the phenotypes of the patients in our study, the paradigm of *de novo* mutational events echoes that observed for other classes of genetic mutation (copy number and single nucleotide variation) deemed pathogenic in ND and CHD cohorts^{9,10}.”

In addition, as detailed below in response to the reviewer’s comments, we have clarified and moderated the presentation of results at multiple points in the manuscript.

Reviewers' comments:

Reviewer #1 (Remarks to the Author):

Barbosa et al revision

This reviewer thanks the authors for a comprehensive rebuttal letter, and accompanying changes to the text that address many comments. Since the rebuttal and changes are quite extensive (and not consistently numbered and tracked), I will attempt to follow the format of the initial rebuttal. I will only mention rebuttal points where I have residual questions.

Reviewer 1

1.1: the authors have answered, but not addressed, the comment about correlating their variations with the clinical presentations of the probands.

As listed above in comments to the Editor, we have added further details that summarize the phenotypes of the study cohort, referencing Supplementary Table 1 that contains a full description of the patient phenotypes. The relevant statement now reads:

“Almost 90% of the patients had a ND, 50% were classified as having an autism spectrum disorder, 16% had an epilepsy/seizure phenotype. 65% also had multiple CAs, including congenital heart defects (CHD) (36%), facial dysmorphisms (29%), growth anomalies (22%), and micro/macrocephaly (13%) (full details in Supplementary Table 1).”

In addition, we have added description of the phenotypes of samples with recurrent epivariations:

Line 190-194: “The two males identified with hypermethylation at *FMR1* both had phenotypes consistent with a diagnosis of fragile X, primarily ID and behavioral anomalies. While both had previously been tested by PCR and reported as normal, subsequent Southern blot testing confirmed the presence of the classical *FMR1* triplet repeat expansion, although in one case this was an apparent mosaic event (data not shown).”

Line 201-203: “One patient with this epivariation at *MOV10L1* presented with double outlet right ventricle, hypoplastic left ventricle, asplenia and short stature, while the second presented with pulmonary stenosis, laryngo-bronchio-tracheomalacia and foot polydactyly.”

1.2a: the statistical criteria are now explained. It is not explicit whether the "stringent" thresholds involved consecutive probes with altered methylation. From other comments in the rebuttal this appears to be so but it would be nice to have this clarified.

The DMR calling script only requires that there are 3 probes either >99.9th percentile or <0.1th percentile of the control distribution within a 1-kb window. However, these do not need to be contiguous/consecutive. Having said that, on manual curation we sometimes removed loci where outlier probes were interspersed with multiple other probes that showed no clear deviation from the norm.

1.2b: this answer deals with the identification of "true positives" based on "manual curation".

(i) Normally I regard a 'true positive' result as one orthogonally demonstrated by a gold standard method - in this case, this should be verification of a methylation change by

bisulphite sequencing. However, the authors use true positive to define a methylation array results after manual curation.

(ii) Manual curation, as well explained here, was chiefly used to weed out results assumed to arise from low signal intensity, and from batch bias. I am quite surprised that the first of these was not achieved during initial QC. The second of these, batch correction, was the subject of my Q2d. I remain a little surprised that these corrections were not applied informatically in initial processing. I am also unsure how any potential bias in interpretation can be applied to manual curation of case and control samples. Were the manual curators blinded to the samples?

In regards to point (i), in the manuscript we use the term “true positive” at only one point (line 120), and this is in reference to the validation rate of DMRs detected by array for which we observed supporting evidence by bisulfite PCR/sequencing.

In regards to point (ii), as explained in Methods, our initial array QC did include removal of probes with failed detection p-values (low signal intensity). However, as with any filtering step, this is never perfect and will also not remove probes that show aberrant signals due to other technical or biological effects (e.g., rare SNPs or indels at the probe binding site or CpG being measured, other technical effects on the array such as wash or hybridization artifacts). We also developed a method to filter regions containing clusters of probes with failed detection p-values, as these were strongly correlated with regions of common CNVs. While this additional step did remove additional probes in regions of poor probe performance, again any such filter will never be perfect.

As explained in my previous response to reviewers’ comments, we did apply standard procedures to reduce batch effects, including quantile normalization of probes (using all cases and controls combined) and application of BMIQ. Again, any such approach is never perfect, and, thus, there will always be some residual batch differences remaining, even after inter-array normalization.

In regards to the final question, no, manual curation was not blinded as to case or control status. However, review was blind to the genomic location, and, thus, there was no bias as to whether specific loci or genes were excluded as false positives during curation in either cases or controls.

1.2c: it's great to have this method clarified. I am still quite surprised that "extreme methylation values" were predicated as outside the furthest control outlier, rather than statistically.

Our criteria for DMR calling required 3 probes in the most extreme 0.1% tails of the control distribution, and thus were in part dictated by the underlying statistics of the controls. In addition, we found that an additional criterion based on magnitude of difference from the control distribution (at least one probe with beta value >0.1 beyond the most extreme value of controls) was required, otherwise at sites where methylation variance was very low, DMRs would be called where the case only showed very small differences compared to controls. The combination of these two criteria for DMR calling had very high specificity, as evidenced by the validation rate by bisulfite PCR of $>90\%$.

1.2d: the rebuttal to my question here re-states some of my remaining uncertainties listed above.

Having looked back at the reviewer’s comments made on the original submission, and those above, I am not quite sure what they are referring to here. In our previous response we attempted to give a full explanation of our data processing pipeline, the rationale behind it, and the evidence

supporting that the DMRs we identify are, by and large, robust. However, if there are further clarifications required on specific points we would be happy to provide them.

1.3b: I still don't like DMR as a term. I guess we agree to differ here.

While we are open to using another term, we are not aware that there is one that is suitable here. Also, as we explained, DMR is in usage in the published literature for many types of methylation change in the genome, and, so, we feel it is an appropriate term to use in the context of our findings.

1.3e, and response figure 2. I was surprised that some of the DMRs presented here are shown to have average methylation difference between 0 and 0.1. I understood from response 2a that the stringent criteria require an average methylation difference ≥ 0.1 .

The reviewer is conflating results from array and bisulfite sequencing. Our DMR calling script (for processing array data) does indeed require that there are three probes, each with difference vs the control mean >0.1 , while the Response Figure 2 shows data from a different technique, *i.e.*, bisulfite sequencing.

I was a bit disconcerted to read that the bisulphite sequencing and 450k results (as presented in Fig 5) are discrepant, and the authors surmise that the 450k results are more valid. As mentioned above (my remark 1.2.b(i)) I believed the gold standard here to be bisulphite sequencing, rather than 450k. It appears that some of the 'large changes' depend on the method used to measure them.

As we explained previously, the results from the array and bisulfite PCR/ sequencing do not always agree, at least if one were to simply consider the summary statistic, *i.e.*, percentage of CpGs measured that are methylated. However, in some instances using this summary statistic alone can lead to erroneous conclusions, particularly in cases where there is likely to be unequal amplification of the methylated and unmethylated alleles.

We illustrate this point in the two figures above, which show the raw data per sequenced molecule for two DMRs which we deemed validated the array findings. In each plot, 50 randomly selected molecules are shown, with each circle corresponding to a single CpG, and each row showing all CpGs tested in a single sequenced molecule: filled circles are methylated CpGs, open circles unmethylated CpGs. Both DMRs were sites where the array reported very low levels of

methylation in controls, and identified a gain of methylation (*i.e.*, hypermethylation, presumably of one allele) in the proband. Results gained by bisulfite PCR/sequencing for both amplicons suggest that the methylation difference between proband and control mean was small (~10%), as measured by the total fraction of methylated cytosines. However, the key point to note here is despite this apparently small change, in both DMRs the proband clearly carries a population of highly methylated molecules that were never observed in any controls.

It was this observation of some highly methylated molecules specifically in the proband (and absent in all controls) that led us to conclude that the bisulfite data validates the initial array finding. While theoretically one would expect that in scenario where a mono-allelic gain of methylation is present, 50% of the sequenced molecules should show a gain of methylation, in reality because PCR reactions will often favor the amplification of one allele over another, biases away from the 50:50 expectation are frequently observed. In these two scenarios, in fact, it appears that the bisulfite PCR is preferentially amplifying the unmethylated allele, such that the methylated allele only represents 5-10% of the sequenced molecules. It is this that causes the final “percent methylation” figure to appear so low in some cases. However, inspection of the raw data in this way clearly shows that the proband with the DMR presumably carries one heavily methylated allele, in agreement with the array findings, despite the relatively small change in total methylation reported by the assay. Thus, the phenomenon of unequal allelic amplification during PCR is what underlies the fact that some loci that we scored as “validated” by PCR to have relatively low percent methylation differences. Of note, a similar phenomenon can also be observed in Figure 1D in the manuscript: here the unmethylated allele is also preferentially amplified (~70% of reads) compared to the methylated allele (~30% of reads).

We hope this explanation provides insight into our validation efforts and reassures the reviewer that we were fair in our assessment of what constitutes secondary validation. While we agree that, in theory, bisulfite sequencing should be an improved “digital technology” compared to arrays, in this case where PCR amplification was used prior to sequencing, this step can lead to significant biases in the final result due to preferential allelic amplification. As such, reliance on only summary statistics can be misleading.

To add clarity to this issue, we have now revised the section of the manuscript where we discuss bisulfite validations to state that in some cases the interpretation of validation experiments was made complex due to highly biased allelic amplification, and have included the above figure as a new Supplementary Figure 3. The section now reads as follows:

Lines 118-127: “Using PCR/bisulfite sequencing, we attempted orthogonal confirmation for 70 epivariations. We observed concordant changes in methylation for 55 of the 58 assays that provided useful data, yielding a 95% true positive rate for DMRs detected by array (Supplementary Table 5). Allelic analysis demonstrated that these epivariations represent large methylation changes specifically on one allele, consistent with the hypothesis that epivariations represent allelic events. In most cases we observed two clusters of largely methylated and unmethylated reads occurring in approximately equal proportions (Fig. 1), although in some instances the interpretation of validation experiments was made complex due to highly biased allelic representation, presumably reflecting preferential PCR amplification of one allele (Supplementary Figure 3).”

1.3.f: The revised text of the authors refers to methylation: transcription correlations from a dataset of 'normal' people. I still feel that to state these 'have an impact comparable to that of loss-of-function coding mutations' is to suggest an inference that is not present in the data. The authors are extrapolating from methylation analysis of patients to expression analysis of different genes in different, healthy, people. It should be very clear that this is what they are doing.

The reviewer is correct that the data presented regarding gene expression in the manuscript are derived from 1000 Genomes samples, not from the cases. This was because, while we could access their DNA, we were unable to obtain RNAseq data from the cases. The section in the manuscript states clearly that expression studies were performed using samples from the 1000 genomes project, but we have now prefaced this with additional wording to state more explicitly that these analyses were done using population controls, and added a clear statement in the conclusion that these observations were made in control samples. The section now reads as follows (new text highlighted in yellow):

Lines 266-277: “In order to provide insight into the biology and functional consequences of epivariations¹⁷, we performed studies of gene expression, inheritance and tissue conservation using datasets of DNA methylation (Supplementary Table 8), gene expression (Supplementary Table 9) and genotype data **derived from population controls**¹⁸⁻²¹. Using paired RNAseq and DNA methylation data in 90 samples from the 1000 Genomes Project, we verified that epivariations encompassing gene promoters were often associated with large changes in gene expression, with hypomethylation leading to increased expression and hypermethylation to transcriptional repression, consistent with the known repressive effects of promoter DNA methylation ($p=9.2 \times 10^{-5}$, Wilcoxon Rank-Sum test) (Fig. 5)²². We also observed that many hypermethylated epivariations at promoters are associated with complete silencing of one allele (Extended Data Fig. 6). **While these observations were made in a control cohort, this suggests that some epivariations** have an impact comparable to that of loss-of-function coding mutations.”

1.4: This revised text is awkward and unclear; it has the feel of a quick fix. It also results in pretty much the first statement of the paper being that methylation changes occur in probands, parents and controls and aren't significant, so that when the original text continues talking about significance, it is a non sequitur.

We apologize if this text was unclear. We have now reorganized and revised the section that discusses the prevalence of epivariations in controls, and the rate of *de novo* DMRs, such that it now reads as follows:

Line 157-177: “**In addition to searching for epivariations in samples with ID/CA, we also screened two large cohorts of population controls, identifying a total of 719 DMRs in the 3,326 control samples analyzed (Supplementary Tables 2 and 3). Thus, epivariations are a relatively common occurrence in the human genome, and are not always associated with any discernable clinical phenotype.** Twenty-four of the epivariations identified in our cases with ID/CA were also found in one or more of these controls, therefore indicating that either that these DMRs are unrelated to the patient phenotype, or perhaps are associated with incomplete penetrance. However, we observed a 1.2 fold enrichment in the frequency of epivariations in the 489 ND/CA samples when compared to 2,711 population controls (Supplementary Fig. 2), although this does not reach statistical significance ($p=0.058$, two-sided Fisher's exact test).

Using a combination of 450k arrays and bisulfite PCR/sequencing assays, we were able to assess the inheritance of 57 DMRs identified in our patients with ID/CA: 33 of the 57 epivariations tested were also present in apparently unaffected parents and, thus, represent inherited events. However, 42% (n=24) of the epivariations identified in patients with ID/CA were absent in both parental samples and, thus, occurred as *de novo* events. When compared to epivariations found in 117 control pedigrees⁷ (Supplementary Table 2), this represents a 2.8-fold enrichment in the rate of *de novo* epivariations in cases compared to controls ($p=0.007$, two-sided Fisher's exact test) (Fig. 2). Thus, while it is currently unclear whether many of the

epivariations identified contribute to the phenotypes of the patients in our study, the paradigm of *de novo* mutational events echoes that observed for other classes of genetic mutation (copy number and single nucleotide variation) deemed pathogenic in ND and CHD cohorts^{9,10}.”

1.5: Two of the three cited examples here (MEG3, FMR1) ARE associated with clinical disorders. They represent positive controls, as it were; and other publications have shown different ways of detecting them. They aren't relevant as stated here.

The statement the reviewer is referring to in the manuscript reads “In addition to their *de novo* nature, recurrence of mutations found in unrelated patients with a similar phenotype is commonly used as a way of assigning significant evidence for the involvement of a specific gene or locus in disease. We identified 12 recurrent epivariations (Extended Data Fig. 3), *i.e.*, the same methylation change was identified in multiple unrelated probands. Of these, two epivariations encompassed the promoters of genes with known disease associations (*MEG3* and *FMR1*)^{6,11}, showing our approach successfully detects pathogenic epivariations.”

We have now edited this to read as follows:

Lines 188-190: “Of these, two epivariations encompassed the promoters of genes **known to show altered methylation in congenital disease** (*MEG3* and *FMR1*)^{6,11}, showing our approach successfully detects pathogenic epivariations.”

As written, we believe this statement is accurate and relevant. We do not make any claim that other methods cannot detect these methylation changes; we are simply stating that we detect things that have previously been shown to be pathogenic. We feel this is an important point, as it supports the entire premise underlying our study that epigenetic profiling can identify pathogenic events in some individuals.

1.6: again the modified text is awkward and adds in clarity "the same regions in other sequenced samples" - how many? which samples? The authors have rebutted, but not quite fixed.

We apologize that this was still unclear, and have modified the text here to be more specific, as follows (edits shown with yellow highlight):

Lines 235-240: “This represents a significant enrichment for rare SNVs disrupting CTCF binding sites in the vicinity of epivariations when compared to the same regions in other **samples in whom we performed targeted sequencing but** who did not carry epivariations at these loci ($p=0.0015$, two-sided Fisher's exact test), strongly implicating rare *cis*-linked variants in regulatory sequence as a causative factor underlying some epivariations.”

Reviewer 2 brings up manual curation and 'epimutation' calling again. In their rebuttal the authors state: 'We deliberately set our DMR calling criteria in such a way as to allow some flexibility to allow for the presence of outlier probes in single controls. If we had not done this, it would otherwise mean that the presence of just one outlier probe found in a single control sample could mean that we could never identify a DMR in that region'. This statement invites the suggestion that statistical criteria might have been more robust.

One can always make criteria more stringent, but, as we had attempted to explain in our initial Response to Reviewers, when we increased our DMR calling thresholds further, we found that we

were no longer detecting DMRs that we knew were real and pathogenic (e.g., *MEG3*), likely indicating that by doing so, we were becoming overly stringent. In the case of controls carrying single outlier probes due to random effects, this represents a genuine problem that cannot simply be solved by increasing the stringency of DMR calling metrics. Specifically, although in any one individual, rare outlier probes might occur for just a few hundred probes per array, when we are using a large control cohort comprising 1,534 samples, this large population size means that at any one locus in the genome there is a good chance that at least one control will have an outlier data point in the region. For an example of this, below we show 450k methylation array data from 4,000 individuals at the imprinted locus *PEG10*. Here, while the vast majority of individuals show intermediate values for all probes, it can be seen at many positions across the locus that there are one or two samples that show beta values that are close to either 0 or 1. Most likely these result from either C>T mutations at the CpG site being measured (C>T mutations via deamination are, by far, the most prevalent mutations in the human genome), or perhaps rare sporadic single base gains or losses of methylation on one allele. As a result of this phenomenon, if we simply used a method that required that a DMR always showed more extreme methylation values than all controls for multiple probes, it would be highly unlikely that DMRs would ever be found at this locus. This realization was what motivated us to require DMRs be defined by three or more probes <0.1th or >99.9th percentile of controls, thus allowing up to 1 per 1,000 controls to have an outlier data point more extreme than in the case. However, our DMR thresholds require three such probes with extreme values within a 1-kb region, at least one of which also has to have a beta value >0.1 beyond the most extreme value observed in all controls. This approach allows DMRs, representing regions of consistent outlier methylation, to be called despite the presence of single rare outlier probes in one sample of the control cohort, and, thus, we contend that this method does add robustness to our approach.

Reviewer 3

3.2: responding to questions about validation, the authors point out that some (10 of 70) methylation changes were not validated by bisulphite sequencing. This invites the comment that the text "we performed orthogonal confirmation for 70 epivariations (Supplementary Table 5), yielding a 95% true positive rate" is disingenuous, since the experiment was attempted on 70 but results secured on only 60.

It is correct that we designed 70 bisulfite PCR assays, although 12 of these failed to yield useful data. The 95% validation rate we quote is based on 55/58 showing a methylation change at the DMR identified by array testing. To clarify these numbers, we have now altered the statement in the text as follows (edits shown in yellow highlight):

Lines 118-121: “Using PCR/bisulfite sequencing, we attempted orthogonal confirmation for 70 epivariations. We observed concordant changes in methylation for 55 of the 58 assays that provided useful data, yielding a 95% true positive rate for DMRs detected by array (Supplementary Table 5).”

3.3: I don't feel the authors directly addressed the comment 'How did the authors test for significance of enrichment in cases vs. controls' in CTCF sites. The response: "This co-localization of DMR and TFBS-variant provides good (albeit circumstantial) evidence of a causal link between the two events" implies no statistical validation.

We apologize if this response was not clear. As stated, we tested for enrichment of CTCF-disrupting mutations using a Fisher's exact test. Although the reviewer is correct in that we sequenced 50 DMR loci, the test was simply asking “did CTCF-disrupting mutations occur more frequently in cases versus controls”, which considers results from all 50 loci together. As we were assessing the overall frequency of CTCF-disrupting mutations associated with DMRs, this represents a single test that includes data from all sequenced loci together, and thus no multiple testing correction is required.

We feel this directly addresses the reviewer's question, which asked “How did the authors test for significance of enrichment in cases vs. controls' in CTCF sites”. We then added additional detail to support this statement by referring to the example of a *de novo* DMR co-localizing with a *de novo* epivariation, but the reviewer is correct in that this latter statement was not accompanied by any statistical test. In this instance we feel this omission is reasonable, as it is inherently unreliable to perform statistics based on a single observation.

3.11 and 3.12 are about MEG3, whose hypomethylation is a recognised association with Temple Syndrome. Given the established diagnostic and research activity for this methylation change, it's a shame the authors didn't secure validation for this assay, though they did present it in fig1.

I am unsure of what the reviewer is requesting here. As stated above, we did attempt validations for 70 loci, which we feel is already a large number, but they are correct that the *MEG3* locus was not among these. Since this *MEG3* DMR had been well described and the subject's phenotype matched that expected for Temple syndrome, we felt this was sufficient. In hindsight, the reviewer is correct that it might have been useful to confirm this DMR with a bisulfite sequencing assay.

3.16: I found this answer startling. How did the authors not previously determine the overlap of methylation changes between cases and controls? It brings me back to my original first comment: that 'the authors have not invested heavily in the clinical characteristics of their cohort or exploring the interplay between molecular and clinical interpretation'. The alterations to the text are a partial fix but I feel there could be more measured moderation of the tone and assertions of the paper.

We would like to make it very clear that our entire DMR calling pipeline operates on the basis of comparing cases versus a large number of controls. Each DMR we defined represented a region where outlier methylation was identified in a case that was not observed in 1,534 control samples.

The edits we made were to then compare the DMRs found in cases to a second control cohort of an additional ~2,600 individuals.

As detailed above, we have now made multiple edits to the text where we moderate and clarify claims, and also add clinical information on the cohort and specific individuals.

Reviewer #2 (Remarks to the Author):

The authors have provided a constructive and thoughtful response to this review. Edits to the manuscript have been made where necessary and I have no further comment.

Reviewer #3 (Remarks to the Author):

The authors have adequately addressed most of my comments. I believe this work presents a strong addition to the research of epigenetic variation as cause for disease and I therefore recommend this paper for publication.

Reviewer #1 (Remarks to the Author):

Barbosa et al, Revision 2.

Many thanks to the authors for taking time and care to develop this thoughtful and comprehensive rebuttal. Most of my comments are adequately addressed.

1. I still feel that the process of identifying DMRs requires informatic and 'manual curation' stages, which are sensitive and potentially subjective, such that an interested reader could not replicate them purely from the information in Supplementary Methods.

The authors have discussed in some detail the decision-making processes for identifying 'true' and 'false' positives and I hope some of this information may find its way into the Methods Supplement as a clear flow of logic.

2. Lines 198-207:

The data on SNV and CNV are presented in supplementary tables 11 and 12 and these should be cited here. Tables S11 and S12 are not self-explanatory:

>270 SNV are presented in Table S12: which are the 7 SNVs regarded as potentially relevant to epigenetic variation?

Which are the 6 CNVs regarded as potentially relevant to epigenetic variation?

13 variants in 50 DMRs seems to me to be 26% not 24%. Is this because one DMR harbours 2 variants? Can this be clarified?

If in cases, 58% of DMRs were apparently inherited, could it be discussed why only 24% apparently showed identifiable genetic changes?

3. Lines 302-316: the analysis of 117 nuclear families produces a smaller proportion of heritability of DMRs than in the case cohort. This should be discussed more. The contention of this paper is that epivariations may be associated with congenital disorders, may be heritable, and may be associated with underlying sequence changes. The discussion on lines 314-316 ('that primary epivariations often exhibit non-Mendelian inheritance') is partly inconsistent with the contention that a significant proportion of epivariations are inherited in the case cohort. If the short discussions of the different cohorts can be drawn together into a consistent position, the overall logic of the paper will be more assured.

Can Tables S11 and S12 be cited at lines 302-303, since these form the data underpinning this statement.

4. Can the authors harmonise two alternative acronyms - ND/CA and ID/CA - which are used at different points in the main text.

5. It seems a bit illogical that for both epigenetic and genetic variations, the data files on controls (Supplementary tables) are cited BEFORE those of cases - eg, Tables S3 and S4 are cited before S2, and S7 long before S11 and S12. This would take only a few minutes to re-draft.

NCOMMS-17-22222B, Identification of rare de novo epigenetic variations in congenital disorders

Response to Reviewer's comments

Reviewer #1 (Remarks to the Author):

Barbosa et al, Revision 2.

Many thanks to the authors for taking time and care to develop this thoughtful and comprehensive rebuttal. Most of my comments are adequately addressed.

1. I still feel that the process of identifying DMRs requires informatic and 'manual curation' stages, which are sensitive and potentially subjective, such that an interested reader could not replicate them purely from the information in Supplementary Methods. The authors have discussed in some detail the decision-making processes for identifying 'true' and 'false' positives and I hope some of this information may find its way into the Methods Supplement as a clear flow of logic.

A more comprehensive explanation of our manual curation has been added to Methods.

2. Lines 198-207:

The data on SNV and CNV are presented in supplementary tables 11 and 12 and these should be cited here. Tables S11 and S12 are not self-explanatory:

These Supplementary Data files have been cited here, and renumbered appropriately, as requested.

>270 SNV are presented in Table S12: which are the 7 SNVs regarded as potentially relevant to epigenetic variation?

These are listed in Supplementary Data 4, Column I. We have added a citation to Supplementary Data 4 at the relevant point in the text.

Which are the 6 CNVs regarded as potentially relevant to epigenetic variation?

These are listed in Supplementary Data 4, Column H. We have added a citation to Supplementary Data 4 at the relevant point in the text.

13 variants in 50 DMRs seems to me to be 26% not 24%. Is this because one DMR harbours 2 variants? Can this be clarified?

The reviewer is correct, Proband103 carried both a CNV and a mutation within a CTCF binding site, as shown in Figures 3 and 4. We have added a note to this effect in the legend of Figure 3.

If in cases, 58% of DMRs were apparently inherited, could it be discussed why only 24% apparently showed identifiable genetic changes?

A summary of our interpretation of the heritability of secondary epivariations is now given in the Discussion.

3. Lines 302-316: the analysis of 117 nuclear families produces a smaller proportion of

heritability of DMRs than in the case cohort. This should be discussed more. The contention of this paper is that epivariations may be associated with congenital disorders, may be heritable, and may be associated with underlying sequence changes. The discussion on lines 314-316 ('that primary epivariations often exhibit non-Mendelian inheritance') is partly inconsistent with the contention that a significant proportion of epivariations are inherited in the case cohort. If the short discussions of the different cohorts can be drawn together into a consistent position, the overall logic of the paper will be more assured.

A summary of our interpretation of the *de novo* generation and heritability of both primary and secondary epivariations is now given in the Discussion. We feel this adequately addresses the reviewer's comment.

Can Tables S11 and S12 be cited at lines 302-303, since these form the data underpinning this statement.

We already cite Figures 3 and 4 here, which we think more concisely support the statement that some epivariations we observed are secondary events related to the presence of an underlying sequence change.

4. Can the authors harmonise two alternative acronyms - ND/CA and ID/CA - which are used at different points in the main text.

This has been done, and we now use ND throughout the manuscript.

5. It seems a bit illogical that for both epigenetic and genetic variations, the data files on controls (Supplementary tables) are cited BEFORE those of cases - eg, Tables S3 and S4 are cited before S2, and S7 long before S11 and S12. This would take only a few minutes to re-draft.

With the earlier citation of Supplementary Data files listing SNVs and CNVs identified in cases, this has now been done for sequence variants. Given the way the Results are written, we believe it is more logical that the numbering of Supplemental Data listing epivariations remains as is.